# The climate change signal in the Mediterranean Sea in a regionally coupled atmosphere-ocean model

Ivan Parras-Berrocal[1], Ruben Vazquez[1], William Cabos[2], Dmitry Sein[3,4], Rafael Mañanes[1], Juan Perez-Sanz[2], Alfredo Izquierdo[1]

[1]Instituto Universitario de Investigación Marina (INMAR), University of Cádiz, Puerto Real, Cádiz, 11510, Spain
[2]Department of Physics and Mathematics, University of Alcala, Alcala de Henares, 28801, Spain
[3]Alfred Wegener Institute for Polar and Marine Research, Bremerhaven, 27570, Germany
[4]Shirshov Institute of Oceanology, Russian Academy of Science, Moscow, Russia

*Correspondence to*: Ivan M. Parras-Berrocal (ivan.parras@uca.es)

**Abstract.** We analyze the climate change signal in the Mediterranean Sea using the regionally coupled model REMO-OASIS-MPIOM (ROM). The ROM oceanic component is global with regionally high horizontal resolution in the Mediterranean Sea, so that the water exchanges with the adjacent North Atlantic and Black Sea are explicitly simulated. Simulations forced by ERA-Interim show an accurate representation of the present Mediterranean climate. Our analysis of the RCP8.5 scenario using the Max-Planck Institute Earth System Model shows that the Mediterranean waters will be warmer and saltier throughout most of the basin by the end of this century. In the upper ocean layer temperature is projected to have a mean increase of 2.7°C, while the mean salinity increases by 0.2 psu, presenting a decreasing trend in the Western Mediterranean, opposite to the rest of the basin. The warming initially takes place at the surface and propagates gradually to deeper layers. Hydrographic changes have an impact on intermediate water characteristics, potentially affecting the Mediterranean Thermohaline Circulation in the future.

## 1 Introduction

The Mediterranean Sea is expected to be among the world's most prominent and vulnerable climate change "hot spots" (Giorgi, 2006; Cramer et al., 2018). As such, the region is an optimal case study site to test new approaches to bridging the gap between science and society, using a sound scientific basis of climate information and applicable to a broad range of vulnerable sectors. The Mediterranean is a regional sea surrounded by Africa, Europe and Asia and divided into two sub-basins (eastern and western) through a sill that does not exceed 400 m depth between Sicily and the African continent. The freshwater balance in the Mediterranean basin is negative, since the evaporation exceeds precipitation and river run-off (Sanchez-Gomez et al., 2011). This deficit is compensated by a net inflow of water through the straits of Gibraltar and Dardanelles. The region is located in a transitional area between tropical and mid-latitudes and presents a complex orography and coastlines where intense air-sea and land-sea interactions take place. These intense air-sea interactions together with the inflow of Atlantic water drive

the Mediterranean thermohaline circulation (MTHC) (Fig. 1), suggesting that atmosphere-ocean regional coupled models (AORCMs) could be conducive to the study of atmospheric and oceanic processes in the Mediterranean Sea.

Different AORCMs with typical horizontal resolution of 25-50 km in the atmosphere and 10-20 km in the ocean have been developed to study the climate of the Mediterranean Sea (Somot et al., 2008; L'Hévéder et al., 2013; Sevault et al., 2014; Cavicchia et al., 2015; Darmaraki et al., 2019). Akhtar et al. (2018) found that higher horizontal resolution (9 km) in the atmosphere improves the simulation of the wind and the turbulent heat fluxes, although they conclude that higher resolution models do not perform better in all aspects than coarser configurations. Somot et al. (2008) developed the Sea Atmosphere Mediterranean Model (SAMM), presenting a new concept of AORCMs through the coupling of the atmospheric global model (ARPEGE; Déqué and Piedelievre, 1995) and the regional high-resolution (10 km) ocean model (OPAMED; Somot et al., 2006). Their results under the A2 (IPCC, 2000) climate change scenario showed an increase of temperature and salinity both in shallow (3.1ºC and 0.48 psu) and in deeper layers (1.5ºC and 0.23 psu) of the Mediterranean Sea (Somot et al., 2006) at the end of the 21$^{st}$ century. In 2013, the European CIRCE project was launched (Gualdi et al., 2013), in order to facilitate the coordination among the scientific community responsible for regional climate modeling in the Mediterranean. The beginnings of CIRCE can be traced back to the work of Dubois et al. (2012) who compared different AORCMs and regional climate models (RCMs). In addition, these authors analyzed a projection (1950-2050) of the Mediterranean climate under the A1B scenario simulated by an ensemble of five coupled regional models. For the first time, realistic atmosphere-ocean net flows were obtained predicting a Mediterranean surface warming between 0.8ºC and 2.0ºC. Shaltout and Omstedt (2014) analyzed the Mediterranean SST for 2005 to 2100 from the Coupled Model Intercomparison Project phase 5(CMIP5) model ensembles under the RCP2.6, RCP4.5, RCP6.0 and RCP8.5 scenarios (Taylor et al., 2012). The CMIP5 ensemble means projected SST warming under all considered scenarios (from 0.5ºC under RCP2.6 to 2.6ºC in RCP8.5). The authors concluded that the warming was mainly controlled by the amount of greenhouse gas emissions. More recently, Adloff et al. (2015) estimated that at the end of the 21st century the mean Mediterranean SST and SSS will increase between 1.73 and 2.97ºC, and 0.48 and 0.89 psu, respectively. Their results were based on an ensemble of six simulations performed with different configurations of the NEMOMED8 (Beuvier et al., 2010) ocean model under different scenarios. Darmaraki et al. (2019) employed an ensemble of 17 fully coupled atmosphere-ocean simulations to study the evolution of SST and marine heat waves in the Mediterranean Sea for the period 1976-2100. The ensemble mean showed a 3.1ºC increase in the Mediterranean mean SST under the RCP8.5 scenario by the end of the century. By 2100 projections showed stronger and more intense Mediterranean marine heat waves. Most of the above-mentioned studies show that the driving factors prescribed in the emissions scenarios condition the expected warming of the Mediterranean Sea.

These modeling efforts are coordinated through the Med-CORDEX initiative (Ruti et al., 2015; www.medcordex.eu), which is the regional climate modeling taskforce of the HyMeX program (www.hymex.org). In the framework of Med-CORDEX a broad range of new reference datasets for regional climate evaluation are being compiled, and the evaluation of new fully coupled regional climate models for understanding the processes that are responsible for the Mediterranean climate variability and trends is being carried out (Somot et al., 2018).

In these models the oceanic component of the AORCMs is also regional. One of the main problems of AORCMs is the prescription of lateral boundary conditions for the regional ocean models, which are mainly based on monthly means from global ocean reanalysis datasets (e. g. HYCOM [Metzger et al., 2014]), damping the ocean dynamics on time scales of less than one month. Those regional climate models should effectively resolve the small-scale processes that are not adequately

represented in the coarser model data used as boundary conditions. This creates inconsistencies between the regional model solution and the external data that can be avoided with the consideration of a global ocean model with refined resolution within the coupled domain (Sein et al., 2015). Such an approach was employed by Izquierdo and Mikolajewicz (2019) in an ocean-only process study to account for the impact of the interaction of processes of different space and time scales on the Mediterranean Water Outflow (MOW) spreading, of particular importance in the Strait of Gibraltar and the Gulf of Cádiz. The

use of an ocean global model (Max-Planck Institute Ocean Model, MPI-OM) in the REMO-OASIS-MPIOM (ROM) coupled system model avoids the problems associated with the open boundary conditions for the Mediterranean Sea, allowing the study of processes that take place in the Mediterranean region but originating in the North Atlantic Ocean. This study aims to contribute to the Med-CORDEX initiative with a first detailed evaluation of high-resolution atmosphere-ocean simulations for present climate with the coupled ROM model. Furthermore, we analyze the evolution of the Mediterranean Sea under the

RCP8.5 scenario with boundary conditions taken from CMIP5 simulation using the Max Planck Institute Earth System model (MPI-ESM). In particular, we focus on ocean properties such as SST and SSS and their evolution towards the end of the 21$^{st}$ century.

The objectives of this study can be summarized as follows:

(i)     Assess the skill of ROM in reproducing the observed Mediterranean Sea regional climate when driven by ERA-Interim

20         reanalysis.

(ii)     Examine the value that high-resolution ROM adds compared to the driving model (MPI-ESM).

(iii)     Assess the projected climate change signal in the Mediterranean Sea under the RCP8.5 scenario.

This article is organized as follows: a general description of our coupled model and each of its components is presented in section 2. In section 3 we present the results of the model validation followed by the coupled simulations for the Mediterranean

region. Finally, section 4 contains the discussion and the conclusions are outlined in section 5.

## 2 Methods

ROM (Sein et al., 2015) comprises the REgional atmosphere MOdel (REMO; Jacob et al., 2001), the Max Planck Institute Ocean Model (MPI-OM; Marsland et al., 2003; Jungclaus et al., 2013), the HAMburg Ocean Carbon Cycle (HAMOCC) model (Maier-Reimer et al., 2005), the Hydrological Discharge (HD) model (Hagemann and Gates, 1998, 2001), the soil model of

REMO (Rechid and Jacob, 2006) and a dynamic/thermodynamic sea ice model (Hibler, 1979) which are coupled via OASIS3.0 (Valcke, 2013) coupler, and abbreviated as ROM from REMO-OASIS-MPIOM.

## 2.1 Atmosphere (REMO)

The atmospheric component of ROM is the REMO. Its dynamic core and discretization in space and time are based on the Europa-Model of the Germany Weather Service (Majewski, 1991). The physical parameterizations are taken from the global climate model ECHAM versions 4 and 5 (Roeckner et al., 1996, 2003). The variables that exchange information between REMO and MPI-OM via OASIS are 10 m wind velocity, wind stress over water, wind stress over sea ice, liquid precipitation, solid precipitation, net shortwave radiation, total heat flux over water, conductive heat flux and residual heat flux (Fig. 2a). To avoid the largely different extensions of the grid cells close to the poles, REMO uses a rotated grid, with the equator of the rotated system in the middle of the model domain. The horizontal discretization is carried out on the Arakawa C-grid and the hybrid vertical coordinates are defined according to Simmons and Burridge (1981). Our version of REMO does not include an aerosol module. The information about aerosols is based on the climatology from Tanre et al. (1984). Here, the spatial distributions of the optical thickness of land, sea, urban, and desert aerosols, and well mixed tropospheric and stratospheric background aerosols are represented. More information about the parameterizations in the atmospheric component can be found in Sein et al. (2015).

## 2.2 Ocean (MPI-OM)

The oceanic component of ROM is the MPI-OM developed at the Max Planck Institute for Meteorology (Hamburg, Germany). MPI-OM is a free surface, primitive equations ocean model, which uses the Boussinesq and incompressibility approximations. MPI-OM is formulated on an orthogonal curvilinear Arakawa C-grid (Arakawa and Lamb, 1977) with variable spatial resolution. This grid allows for the placement of the poles over land, thus removing the numerical singularity associated with the convergence of meridians at the geographical North Pole. An additional advantage of the curvilinear grid is that a higher resolution in the region of interest can be obtained, while maintaining a global domain. Using the global ocean model alleviates issues related to ocean open boundary conditions and provides an additional "degree of freedom" in the model setup and tuning, which can help increase the performance of the ocean component within the region of interest. The model parameterizations and setup are described in Sein et al. (2015).

## 2.3 ROM configuration and experiments set-up

Fig. 2a shows the coupling scheme used in ROM. In the region covered by REMO the atmosphere and the ocean interact while the rest of the global ocean is driven by energy fluxes, momentum and mass from global atmospheric data used as external forcing. In the experiments analyzed here, data from ERA-Interim reanalysis (Dee et al., 2011) and MPI-ESM-LR (Giorgetta et al., 2013) are used to provide lateral boundary conditions to REMO and to force MPI-OM outside the coupling region. The MPI-OM grid used in this setup is represented by black lines in Fig. 2b. In the Mediterranean region the highest horizontal resolution of MPI-OM is 7 km (south of the Alboran Sea) while the lowest resolution is 25 km (eastern coasts of the Mediterranean Sea). MPI-OM has 40 vertical z-levels with increasing layer thickness with depth, with the first layer nominal

thickness of 16 m. The spin-up of MPI-OM was done according to the procedure described in Sein et al. (2015): In the stand-alone mode, MPI-OM is started with climatological temperature and salinity data (Levitus et al., 1998). Subsequently, it is integrated four times through the 1958-2002 period forced by ERA40. For the coupled runs, the model is started from the final state reached in the last stand-alone run and integrated again two times forced by ERA-40 and one time forced by ERA-Interim

reanalysis (1979-2012).

The REMO domain covers the North and Tropical Atlantic, a large part of Africa, South America and the Mediterranean region (red line, Fig. 2b) with a resolution of approximately 25 km on a rotated grid and a time step of 120 s. More information about the ROM coupled system is summarised in Table 1. The HD model (global domain) computes the river discharge at 0.5° resolution and an information exchange takes place every 60 minutes, while HD interacts with MPI-OM and REMO every 24

hours (Fig. 2a).

In this study, 30-year time series from three different experiments have been analyzed. The first simulation, ROM_P0, was forced by ERA-Interim for the time period 1980-2012 and used to assess the skill of ROM to reproduce the observed regional climate over the Mediterranean Sea. In order to present an integrated vision of the impact of climate change in the Mediterranean Sea, we dynamically downscale the MPI-ESM-LR historical simulation covering the period 1950-2005 (for

our analysis we take ROM_P1 from 1976-2005) and the climate change projection for 2006-2099 (for our analysis we take ROM_P2 from 2070-2099) under the Representative Concentration Pathway 8.5 (RCP8.5) scenario.

The driving model, MPI-ESM, has been used in different configurations for CMIP5 in a series of climate change experiments (Giorgetta et al., 2013). MPI-ESM is composed of ECHAM 6 (Stevens et al., 2013) for atmosphere and MPI-OM (Jungclaus et al. 2013) for ocean as well as JSBACH (Reick et al., 2013) for terrestrial biosphere and HAMOCC (Ilyina et al., 2013) for

the ocean's biogeochemistry. The coupling of the atmosphere, ocean and land surface is made possible by the OASIS3 (Valcke, 2013) coupler. MPI-ESM-LR (low resolution) uses T63/1.9° horizontal resolution and 47 hybrid sigma/pressure levels for the atmosphere and a bipolar grid with 1.5° horizontal resolution (near the equator) for the ocean, while the -MR (mixed resolution) version has the same horizontal resolution in the atmosphere, although doubles the number of vertical levels in the atmosphere and decreases the horizontal grid spacing of the ocean to 0.4° by means of a tripolar grid (Giorgetta et al., 2013).

We have used MPI-ESM-LR to force ROM in experiments ROM_P1 and ROM_P2 because -LR was used in a wider set of CMIP5 experiments and with more realizations than -MR (Giorgetta et al., 2013). Both present the same horizontal resolution in the atmosphere, and although MPI-ESM-MR has a higher vertical resolution, mainly in the upper troposphere and lower stratosphere, the main differences in the simulations can be found in the middle atmosphere (Stevens et al, 2013). According to a recent benchmarking exercise of CMIP5 models (Lauer et al., 2017) their overall performance is quite similar. Jungclaus

et al. (2013) provided a detailed description and evaluation of the ocean performance of MPI-ESM-LR and -MR, and concluded that both behave similarly in many aspects, although -LR simulated the Labrador Sea and the North Atlantic more accurately, at least in the mean state and its variability.

## 2.4 Validation Methodology

ROM-simulated present Mediterranean climate is analyzed in terms of mean state, seasonal cycle and interannual variability of several atmospheric and oceanic variables. For the ROM atmospheric component REMO, three representative variables were chosen: Mean Sea Level Pressure (MSLP), near-surface temperature (T2m) and total precipitation. For the ocean component MPI-OM Sea Surface Temperature (SST), Sea Surface Salinity (SSS), Sea Surface Height (SSH) and the sub-surface current velocity are considered. These fields are compared to gridded data from different sources (interpolated observed data and reanalysis) to evaluate the ROM's ability to simulate the present Mediterranean climate (Table 2).

For MSLP and T2m we compare the output of ROM with ERA-Interim reanalysis. The ERA-Interim data assimilation system uses a 2006 release of the Integrated Forecasting System (IFC) developed jointly by ECMWF and Météo-France. The spatial resolution of the dataset is approximately 80 km (T255 spectral) on 60 vertical levels from the surface up to 0.1 hPa (Dee et al., 2011); data can be freely accessed at https://www.ecmwf.int/en/research/climate-reanalysis/era-interim. Total precipitation is validated against the Tropical Rainfall Measuring Mission (TRMM; Huffman et al., 2014) dataset, a joint mission between NASA and the Japan Aerospace Exploration Agency (JAXA) to study rainfall for weather and climate research.

Three datasets were used for the evaluation of the SST: ERA-Interim, EN4 and OISST. EN4 was derived by Good et al. (2013) who carried out a 1-degree monthly objective analysis from ocean temperature and salinity bathythermograph profiles (MBT, XBT). The version EN4.1.1 used here includes the improvements of the estimation of MBT's and XBT's downward velocity developed by Gouretski and Reseghetti (2010). The NOAA daily Optimum Interpolation Sea Surface Temperature version 2 (OISST; Reynolds et al., 2007) combines observations from different platforms (satellites, ships, buoys) on a regular global grid 1/4º x 1/4º. The OISST dataset offers an accurate representation of sea surface (Ferster et al. 2018) and is widely used in the evaluation of regional climate models (e.g. L'Hévéder et al. 2013, Akhtar et al. 2019, Cabos et al. 2019).

For SSS we used the two following datasets: EN4 v.4.1.1 (Good et al., 2013) and MEDSEA_REANALYSIS_PHY_006_009 (Fratianni et al., 2015) implemented by the Copernicus Marine Environment Monitoring Service (CMEMS) and with a 1/16º horizontal resolution in the Mediterranean.

The potential of ROM to improve the simulation of the regional Mediterranean Sea climate is assessed by comparisons against the MPI-ESM outputs (MPI-ESM-LR and MPI-ESM-MR).

## 3 Results

In this section, a selection of key fields corresponding to the period 1980-2012 of ROM forced by ERA-Interim (ROM_P0) is presented. In a second step changes in the Mediterranean Sea under RCP8.5 conditions are estimated from the analysis of differences between present climate (1976-2005, ROM_P1) and the climate projection (2070-2099, ROM_P2) carried out by ROM driven by MPI-ESM-LR.

## 3.1 Atmosphere validation

Mean sea level pressure (MSLP) is a good indicator of large-scale circulation, which influences near-surface temperature (T2m) and precipitation distributions. Erroneous MSLP gradients lead to an erroneous regional wind circulation, and can also have a strong effect on ocean circulation (Sein et al., 2015). Figs. 3a and 3b display the biases of modeled MSLP with respect to ERA-Interim for the boreal winter (defined as December, January, and February; DJF) and summer (defined as June, July, and August; JJA) in the 1980-2012 period (ROM_P0). ROM_P0 provides a good agreement with ERA-Interim MSLP, showing maximum deviations smaller than 3 hPa over most of the domain for both seasons. The strongest departures can be found in DJF, due to an overestimation of the Azores high during the winter months. Those differences could be attributed partly to REMO parameterizations, but a more important role could be played by the deficiencies in the simulated ocean circulation in the North Atlantic, which result in a region of cold SST bias centered east of Flemish Cap (not shown). Jungclaus et al. (2013) consider this cold bias appearing in MPI-ESM-LR and -MR to be a persistent feature in state-of-the-art climate models, where the coarse resolution prevents a proper representation of the Gulf Stream separation (Dengg et al., 1996), although they also mention other possible causes. Nonetheless, these relatively small deviations imply a small change in terms of regional wind circulation. During summer months (Fig. 3b) MSLP biases are much smaller over the Mediterranean.

Figs. 3c and 3d show T2m biases for DJF and JJA. For both seasons the departures are typically below 3.0ºC over most of the coupled domain, except for the Alps, the Pyrenees, the Atlas, the Caucasus and the Armenian highlands (Figs. 3c and 3d). This disagreement can be attributed to differences in the resolution of orographic features. Winter months show the largest T2m biases located close to the Mediterranean coastline.

At first glance, ROM_P0 generally underestimates the simulated cumulative precipitation over most of the Mediterranean region, for both winter and summer seasons. The largest discrepancies for DJF are located over the Black Sea, the Adriatic Sea and the Gulf of Lions (Fig. 3e), where negative anomalies can reach 3mm/d. Moreover, it is worth stating that during the same period the total precipitation was overestimated in regions linked to significant topographic reliefs (e.g. the Alps). Some coastal areas also showed positive anomalies that are most likely related to the transport of precipitable water, which is influenced by the simulated evaporation over the ocean (atmosphere-ocean coupling). In the very dry Mediterranean summer season, ROM_P0 shows a clear tendency to underestimate the precipitation (Fig. 3f). Over the ocean, this bias can be related to the cold SST bias, common to most of the AORCMs simulations of the Mediterranean climate (see Darmaraki et al., 2019). The seasonal mean precipitation is reasonably well simulated by our coupled system throughout most of the Mediterranean basin. However, the systematic errors (up to ±3.5 mm/d) remain substantial over the region in terms of precipitation.

The impact of interactive atmosphere-ocean coupling in REMO is shown in Fig. 4, presenting the climatology differences between ROM_P0 and stand-alone REMO in the simulations forced by ERA-Interim for MSLP, T2m, and precipitation. Over land the simulated fields are less influenced by the coupling and are largely dependent on the details of the atmospheric component. On the other hand, the impact of the coupling can be remote, through the large-scale circulation (the signal which comes from the North Atlantic) and the land-sea contrasts account for the local effects. Therefore, we can expect the differences

over land to be minimal, except for the regions where the large-scale circulation or the land-sea contrasts are significant. In addition, the ROM model uses an orographic gravity wave drag formulation that improves the representation of the circulation over mountainous regions in REMO.

The winter MSLP over the Atlantic is higher in the coupled run (Fig. 4a), causing an anomalous strong anticyclonic circulation that extends to land and the Mediterranean Sea, west of the Balearic Islands. The influence of the large-scale MLSP anomaly cancels the effect of the local warmer SST, which would create a low-pressure bias here (see Fig. 5, where the SST biases are represented). However, elsewhere over the Mediterranean Sea, where the ROM_P0 SST is colder (warmer) than ERA-Interim, a higher (lower) MSLP is simulated by ROM_P0. In summer (Fig. 4b), the differences in MSLP seem to be determined mainly by the colder SST in ROM_P0, which leads to higher MSLP in the model than in the reanalysis.

The changes in T2m induced by the coupling over the Mediterranean (Figs. 4c and 4d) seem to be determined mainly by the SST (see also Fig. 5), through the turbulent heat fluxes. In both seasons the T2m differences induced by the coupling correspond very well with the SST biases with respect to ERA-Interim. However, in winter T2m also seems to be influenced by the transport of Atlantic air carried by the too strong anticyclonic circulation simulated in the Atlantic. Over land the differences in winter T2m are mainly determined by the changes induced in large scale circulation by the interactive SST in the Atlantic, while in summer the land-sea contrasts seem to be more significant.

The differences between the SST from ERA-Interim and the simulations by ROM_P0 are also reflected in the rainfall simulated by REMO and ROM_P0 (Figs. 4e and 4f) as shown by the correlation (r=0.63) between winter SST and precipitation biases (including the Black Sea). In winter, the Mediterranean Sea regions where the ROM_P0 SST is warmer have higher precipitation, while colder ROM_P0 SST leads to lower precipitation. The prevalent summer cold SST bias in ROM_P0 leads to weaker precipitation throughout the Mediterranean Sea, especially in the northern part.

## 3.2. SST

### 3.2.1 Seasonal cycle

The differences between ROM_P0 and observed SST climatology for winter (DJF) and summer (JJA) in the period 1980-2012 are presented in Fig. 5. The SST seasonal cycle is well represented by the model, although its amplitude is reduced over most of the Mediterranean Sea. The deviations in absolute value do not exceed 3.0ºC, although ROM_P0 shows a cold bias, which is more significant in the northern part of the eastern Mediterranean Sea, especially in summer (Fig. 5).

In DJF ROM_P0 overestimates SST over the northern Mediterranean coasts and the whole western basin, showing warm biases reaching 2.0ºC (Figs. 5a, 5b and 5c). In summer, the cold SST bias extends over a large part of the Mediterranean domain (Figs. 5d, 5e, and 5f).

In order to assess the improvement that higher resolution in ROM brings to the simulation of the present Mediterranean climate (ROM_P0), comparisons with MPI-ESM-LR and -MR have been done (Fig. 6):

SST seasonal cycle amplitude is smaller in ROM_P0 than in MPI-ESMs, with warmer DJF and colder JJA. The SST differences are smaller than 3.0ºC in the whole Mediterranean basin. In winter, ROM_P0 shows warmer temperatures than MPI-ESM (-LR and -MR, Figs. 6a and 6b) with the exception of southeastern Mediterranean coasts where negative differences appear (approximately -1.0ºC). In JJA ROM_P0 is significantly colder over the western basin (-1.5ºC), southern coasts (-0.5ºC), Levantine and Aegean seas (-3.0ºC) while it is warmer in the Tyrrhenian, Adriatic and Ionian seas (up to +1.0ºC; Figs. 6c and 6d).

### 3.2.2 Interannual variability

The time series of yearly mean SST averaged over the Mediterranean Sea for the period 1980-2012 (ROM_P0) shows cold biases (from 0.1 to 1.4ºC) against ERA-Interim, EN4 and OISST datasets (Fig. 7), in agreement with the results displayed in Fig. 5. ERA-Interim (purple line) and OISST (red line) present a consistent behavior and ROM_P0 shows a mean cold bias of 0.6ºC. The largest deviations are found for EN4 (yellow line) due to the lower resolution of the dataset.

ROM_P0 shows a warming trend in SST, as in the observational datasets, albeit slightly weaker (Table 3). Also, the interannual variability evident in the observed datasets is properly reproduced by ROM_P0.

A Taylor diagram (Fig. 8) was used to quantitatively evaluate ROM_P0 performance. ERA-Interim, EN4 and ROM_P0 are all well correlated (r>0.7) with the observation-based analysis (OISST). ROM_P0 SST standard deviation (0.27ºC) is close to that of OISST, ERA-Interim and EN4 (0.32, 0.34 and 0.33ºC, respectively). The corresponding root-mean-square-errors (RMSE, red contours) show good ROM_P0 performance simulating the interannual variability of SST, with ROM_P0 closer to EN4 than EN4 to OISST and ERA-Interim. This could be interpreted as ROM_P0 SST lying outside but close to the uncertainty range inherent to observational gridded datasets.

### 3.3 SSS

Fig. 9 shows the differences between the SSS modeled by ROM_P0 and the selected datasets averaged for DJF and JJA during the period 1980-2012. All cases show a positive bias over the western basin and Adriatic Sea and negative bias throughout the Levantine Sea and north Aegean Sea. In the northeast Adriatic Sea, by the Po Delta, the largest positive differences occur (3.0 psu), and to the north of the Aegean Sea the largest negative differences (-3.0 psu) are found. Nevertheless, the deviations do not exceed, in absolute value, 0.5 psu in a large part of the domain (Fig. 9). Deficiencies in simulated precipitation are propagated into HD model river discharge, which is reflected in the SSS. ROM simulated total river runoff into the Mediterranean is smaller than most of the observational estimates (e.g. Struglia et al., 2004; Wang and Polcher, 2019) and lower than other AORCMs estimates (see Table 4). The influence of river runoff on SSS is highlighted by the coincidence of largest SSS biases with locations of large rivers (Po, Nile) and with the Dardanelles Strait, whose net flow is larger than estimates (Sánchez-Gómez et al., 2011).

The ROM_P0 SSS is compared with MPI-ESM-LR and -MR in Fig. 10. ROM_P0 is always saltier over the whole Mediterranean, with a decreasing difference towards the southeast. In general, ROM_P0 SSS is closer to EN4 and CMEMS climatologies than any of the MPI-ESM versions, due to the higher horizontal resolution of ROM_P0 in atmosphere and ocean.

### 3.4 SSH and circulation

To conclude with the analysis of the ocean component of ROM, the SSH was analyzed. The time-averaged SSH and horizontal current velocity at 31 m depth simulated by ROM_P0 between 1980 and 2012 are shown in Fig. 11. The 31 m depth level has been chosen to remove the high-frequency variability of the uppermost ocean, while retaining a characteristic upper ocean circulation pattern. Furthermore, the choice of this depth makes our results more comparable with previous studies, such as L'Hévéder et al. (2013) and Sevault et al. (2014). It can be clearly seen that Atlantic surface waters enter through the Strait of
Gibraltar to the Western Mediterranean; after crossing the Alboran Sea the Atlantic water flows along the African coast. At the Strait of Sicily, part of the Atlantic water deflects northward along the coast of the Tyrrhenian Sea, while the rest continues flowing to the Eastern basin. ROM_P0 reproduces quite clearly the well-known deep water formation sites, especially in the Gulf of Lions, southern Adriatic Sea and in the Levantine Sea (near Crete and Rhodes islands) identified by the presence of three cyclonic gyres. These cyclonic gyres concur with negative SSH values, which highlight the sinking of surface waters.
The mean SSH closely reproduces the well-established steady basin and sub-basin scale circulation pattern (e.g. Bergamasco and Malanotte-Rizzoli, 2010). However, meso-scale structures of circulation such as Mersa Matruh and Shikmona anticyclonic gyres escape the model's horizontal resolution in the Eastern basin (ca. 25 km).

A first order comparison of the model's SSH to the AVISO Sea Level Anomaly (SLA) (SSALTO/DUACS, 2013) can be done by adding only the thermosteric contribution (as a constant resulting from the average over the whole basin) to the dynamic
SSH of the model (Sevault et al., 2014). Fig. 12 shows the yearly mean and the seasonal cycle of ROM_P0 SSH compared to altimetric data. The modeled SSH shows lower values than the observed (Fig. 12a); however, it represents well the behavior of the AVISO SLA time series. The amplitude of the mean seasonal cycle is 12 cm for the simulation, and 14.5 cm for AVISO (Fig. 12b). Therefore, the model is able to reproduce a realistic interannual variability and seasonal cycle.

Finally, a mass balance was carried out to estimate the net transport of water throughout the Strait of Gibraltar and Dardanelles
in order to compare the water flux modeled by ROM with the observations. Table 4 gives the water budget of ROM_P0 averaged over the period 1980-2012. The water loss by evaporation (E) is greater than the gain by precipitation (P) and river runoff (R) generating a deficit of 0.053 Sv in the basin. However, this deficit is partially compensated by the net water inflow through the Strait of Gibraltar (0.030 Sv) and the Dardanelles, where the inflow (0.132 Sv) exceeds the outflow (0.109 Sv). ROM_P0 water budget (E-(P+R)) is 0.007 Sv lower compared to the RCSM4 model (Sevault et al. 2014), although a significant
part of the difference is due to difference in river runoff.

## 3.5 Projections under RCP8.5 scenario

Fig. 13 shows the mean SST and SSS fields for the present climate (1976-2005, ROM_P1) together with the differences with respect to future projections under the RCP8.5 scenario (ROM_P2-ROM_P1). At a basin scale, the SST (ROM_P1, Fig. 13a) increases from northwest to southeast over the Mediterranean Sea, with the western Mediterranean colder than the eastern, especially in the Gulf of Lions and in the northern Adriatic Sea where the SST minima are located (Fig. 13a). The warmest area is found along the Levantine Sea coast. The averaged Mediterranean SST is 18.6ºC and at the end of the 21st century under RCP8.5 scenario it is expected to have a mean increase of 2.7ºC, with a projected warming ranging from a maximum of 3.8ºC at the Aegean Sea to a minimum of 0.9ºC at the Alboran Sea (Fig. 13b).

To verify that the simulated warming trend remains stable and is not affected by the strong ROM SST bias, comparisons for DJF and JJA have been performed separately (see Supplementary Figures). The comparable warming is appreciable in both seasons, with a larger SST in the eastern basin. The influence of seasonal cycle is limited to the location of the minima and maxima.

As shown in Fig. 13c, at the surface the Eastern Mediterranean is saltier than the Western Mediterranean, reaching 39.0 psu at the Levantine Sea. The Western basin presents lower salinities (< 38.3 psu) influenced by the inflow of less saline Atlantic water through the Strait of Gibraltar (36.6 psu) along the African coasts up to the Ionian Sea. Another source of freshwater is located at the Dardanelles strait where the Black Sea outflow has salinities lower than 35 psu. The averaged Mediterranean SSS is 38.0 psu, while under the RCP8.5 projection it will experience a mean increase of 0.2 psu. The differences between the mean SSS projection and present climate shows a dipolar structure through the Mediterranean Sea (Fig. 13d). Under the RCP8.5 scenario, the Western Mediterranean is expected to increase slightly in fresh water (from -0.5 to -1.0 psu), while the Eastern will become saltier. It is precisely at the north of the Aegean Sea where largest SSS increases (4.0 psu) are found.

MPI-ESM-LR and -MR projections under the RCP8.5 scenario at the end of the 21st century are slightly warmer than that of ROM over most of the Mediterranean Sea. Namely, the projected mean SST increases are 2.8 and 2.9ºC for MPI-ESM-LR and -MR, respectively (Table 6). Compared to ROM, both MPI-ESMs show a tendency to shift the largest warming to the west, more notoriously in MPI-ESM-MR, with a local minimum extending over the eastern basin (Figs. 14a and 14b). It is also remarkable that MPI-ESM-MR identifies the maximum warming at the Adriatic Sea and the northern Aegean Sea in the Dardanelles water outlet (Fig. 14b).

The mean SSS increase projected by ROM for the 2070-2099 period compared to 1976-2005 under RCP8.5 (ROM_P2-ROM_P1) is larger than for any of the MPI-ESMs (Table 4), but the salinity change dipolar spatial pattern is roughly the same in all three projections (Figs. 13d, 14c and 14d).

Fig. 15 shows the mean temporal evolution of temperature and salinity anomalies in the water column over the Western and Eastern Mediterranean throughout the 21st century according to ROM projection for the RCP8.5 scenario. To calculate these anomalies in a given region we first average horizontally, over the area indicated in Fig. 15 insets, the temperature and salinity in each MPI-OM level for the present time period (1976-2005) and the RCP8.5 projection period (2006-2099). The anomalies

are defined as the difference between the time series for the RCP8.5 scenario (2006-2099) and the time mean for the present climate period (ROM_P1). The Mediterranean Sea shows a gradual increase of its temperature throughout the entire water column (Figs. 15a and 15c), which is most pronounced in surface layers. The warming accelerates in the second half of the century, with a very clear warming signal in the upper 500 m in the Eastern Mediterranean. This warming signal propagates at intermediate depths (200-500 m, corresponding to the equilibrium depth of Levantine Intermediate Waters (LIW), e.g. Menna and Poulain, 2010) into the western basin. At the end of the 21st century the eastern basin is expected to experience a surface temperature increase of up to 3.8ºC and the western up to 3ºC. At 1000 m depth the water temperature will increase by 0.6ºC for both basins, which is a very notable warming at these depths.

The time evolution of mean salinity anomalies displays different patterns throughout the Mediterranean Sea. During the 21st century the upper layer (0-100 m) of the Western Mediterranean is projected to freshen (-0.5 psu) while the deeper layers tend to get saltier up to 0.5 psu. However, the Eastern Mediterranean will increase its salinity up to 0.5 psu in the entire water column. It is interesting to note that both temperature and salinity increases in the Western Mediterranean at intermediate depths are delayed compared to the eastern Mediterranean.

## 4 Discussion

AORCMs are capable of improving the simulation of the climate system by the driving model through dynamical downscaling from GCMs (e.g. Li et al., 2012; Sein et al., 2015). The regionalization implemented in ROM model provides higher horizontal resolution, allowing the representation of local scale and mesoscale processes that are not detectable by MPI-ESMs. The higher horizontal resolution also allows ROM_P0 to resolve explicitly the water exchange through a more realistic Gibraltar and Dardanelles Straits, taking into account the large-scale feedbacks between the Mediterranean and the adjacent basins (North Atlantic and Black Sea) Compared to other state-of-the-art regional climate models, ROM introduces the novel approach of implementing a global ocean model with high horizontal resolution at regional scales. This allows us to obtain information of the global ocean maintaining the high spatial resolution in the coupling area. An important disadvantage of the proposed model, described previously in Sein et al. (2014), is that the bias and internal variability generated from the global domain can influence the results in the coupled domain, making it difficult to separate the source of bias.

ROM is able to reproduce the main characteristics of the climate of the Mediterranean Sea. The biases of the main atmospheric and oceanic parameters are in the range shown by other state-of-the-art regional models (L'Hévéder et al., 2013; Sevault et al., 2014; Akhtar et al., 2018; Darmaraki et al., 2019).

The seasonal MSLP was validated against ERA-Interim, showing biases smaller than ±3 hPa over most the domain for DJF and JJA, a performance similar to other models (see e.g. Giorgi and Lionello, 2008; Velikou et al., 2019). Positive MSLP biases over a large extent of the domain during DJF (Fig. 3a) could generate anticyclonic conditions which lead to a greater stability and lower storm generation; while in JJA (Fig. 3b) the biases are generally much lower. With respect to the seasonal cycle of near-surface atmospheric parameters such as near-surface (2m) temperature (T2m) and precipitations, the LMDz-

NEMO-Med coupled model (L'Hévéder et al., 2013) (Table 5) gives a bias (range of -4; +4°C/-2; +3 mm/d, respectively) which is comparable to the ROM_P0 estimates (Figs. 3c, 3d, 3e and 3f). Similar to most of the Mediterranean regional models, ROM_P0 shows higher than observed rainfall over areas with pronounced topography, such as the Alps (Artale et al., 2010; L'Hévéder et al., 2013; Di Luca et al., 2014) (Table 5). More recently, Fantini et al. (2016) also reported a similar bias (±3

mm/d) in an ensemble of regional coupled models forced by ERA-Interim. Panthou et al. (2018) observed that for heavy precipitation increasing resolution increases the wet biases when comparing simulations that share the same set of parameters. We agree with the final consideration of Fantini et al. (2016); the authors propose that in order to assess the performance of the RCMs with ever increasing resolution in simulating precipitation, we urgently need observations with high temporal and spatial resolution.

The comparison of the ROM_P0 with stand-alone REMO shows that the changes in SST generated by the coupling in the Atlantic Ocean influence the simulated Mediterranean climate, causing a spurious anticyclonic circulation in winter which impacts the surface temperature in the Western Mediterranean. In summer the modeled SST is significantly colder than observations, leading to colder T2m and less precipitation over the basin, as the colder SST reduces the evaporation. In order to explicitly assess the role of the regional coupling on the simulated temperature, salinity and sea level, the results presented

here will be compared with those from an uncoupled MPI-OM simulation, which is in progress.

Regarding SST ROM_P0 shows biases within 3.0°C, correlation coefficients above 0.7 and RMSE below 0.25°C when compared to ERA-Interim, EN4 and OISST datasets. ROM_P0 presents cold biases along the Eastern Mediterranean that become stronger and extend to the whole basin in summer months. The summer biases are common to most of the Mediterranean regional coupled simulations (see for instance, Dubois et al., 2012; Li et al., 2012, Sevault et al., 2014). Akhtar

et al. (2018) studied the impact of resolution and coupling in modelling the climate of the Mediterranean Sea and concluded that coupling generates a negative bias in SST. Most recently, Darmaraki et al. (2019) assessed an ensemble of 17 simulations from six models, in which our ROM coupled system was included. Their results showed an averaged cold bias ranging from (-0.29 to -1.01°C) when regional models are compared to satellite data. This cold bias is very evident in Fig. 7, where ROM_P0 shows averaged Mediterranean SSTs that reproduce the trend and interannual variability but are systematically colder than

reference climatologies during the period 1980-2012, a common trait with other RCSMs (Sevault et al., 2014; Ruti et al., 2015). Macias et al. (2018) showed that a simple spatially-uniform bias correction improves the simulated surface oceanic conditions of the Mediterranean basin when forcing an oceanic model with atmospheric data from RCM realizations. The causes of the cold summer SST biases could be related either to a deficit of solar radiation by the atmospheric model or to shortcomings in the simulation of certain processes in the ocean model, such as vertical mixing or turbidity. It is difficult to

attribute the bias to single cause without considering the multiplicity and complexity of all the involved conditions; therefore, this topic deserves a separate and focussed study. However, a preliminary sensitivity analysis (not shown) changing the optical properties of the water (changing from model standard Jerlov Ia to Jerlov II) clearly indicates that the related turbidity increase is responsible for a larger absorption of downward shortwave radiation in the upper layer, and leading to a warmer SST. This also would explain why colder SST biases in summer, when the impact of biologically-induced redistribution of heat in the

water column is larger. Switching HAMOCC on could, to a certain extent, contribute to the reduction of this cold bias. However, until a thorough study is carried out, the contribution of other mechanisms cannot be discarded. The SSS simulated by ROM_P0 shows seasonal biases within 1 psu, with a similar magnitude and spatial distribution to those in RCSM4 (Sevault et al., 2014). The biases are higher in areas such as the northern Adriatic Sea and Dardanelles Strait (Fig. 9), a feature that has

also been shown in previous studies (L'Hévéder et al., 2013; Di Luca et al., 2014; Sevault et al., 2014). The Mediterranean water fluxes simulated by ROM_P0 (Table 4) have been compared to available observations (Sanchez-Gomez et al., 2011; Soto-Navarro et al., 2014) and model (Sevault et al., 2014) estimates, providing a physically consistent assessment in the straits. ROM_P0 water balance terms over the Mediterranean Sea are similar to those obtained by different authors (Table 4). The main difference is the exchange flows through the Strait of Gibraltar, where ROM_P0 presents estimates much lower than

those shown by Soto-Navarro et al. (2014), although the net flow is in agreement with most estimates.

The ROM_P0 SSH and surface (31m) circulation are able to reproduce the different stationary elevation/depression (anticyclonic/cyclonic) structures occurring in the Mediterranean Sea (Fig. 11). The cyclonic gyres (SSH depressions) correspond to the water mass formation sites. For the period 1980-2012 the comparison between ROM_P0 and AVISO (SSALTO/DUACS, 2013) altimetry data (Fig. 12a) produced a satisfactory correlation of 0.61, similar to that obtained by the

RCSM4 (0.68) (Sevault et al., 2014). Finally, the ROM_P0 amplitude of the mean seasonal cycle measured was 12 cm while for AVISO it was 14.5 cm (Fig. 12b) and for RCSM4 16.9 cm (Sevault et al., 2014).

In general, despite some systematic errors, we have shown that ROM_P0 satisfactorily reproduces the mean state, seasonal cycle and interannual variability shown in the analyzed variables from ERA-Interim (1980-2012). There is a clear improvement over the driving MPI-ESM, and ROM_P0 skills are comparable to other AORCMs. The use of a global ocean grid allows us

to overcome the difficult prescription of ocean lateral boundary conditions, but more importantly, to take into account the possible feedbacks between changes in Mediterranean Sea state and changes in the adjacent North-Atlantic and Black Sea, which may be of importance for climate projections, by means of an explicit exchange through the Straits of Gibraltar and Dardanelles. Adloff et al. (2015) studied the Mediterranean Sea response to climate change by means of a set of numerical experiments using the regional ocean model NEMOMED8 and concluded that the sensitivity of the evolution of the

Mediterranean water masses to the choice of the Atlantic boundary conditions is at least of the same order as the sensitivity to the choice of the socio-economic scenario. The model also proved capable of reproducing the area-averaged interannual standard deviations of SST for the Mediterranean Sea (Fig. 16d). As seen in Fig. 16, the ROM coupled system presents yearly SST standard deviations close to the reference OISST dataset. In fact, ROM_P0 does not only improve the yearly spatial standard deviations compared to the MPI-ESMs (Figs. 16e and 16f) but also compared to ERA-Interim and EN4 (Fig. 16b and

16c). The MPI-ESM-LR and -MR are not able to reproduce those local patterns due to the coarse resolution, which indicates that the dynamical downscaling from MPI-ESM refines the fields simulated by the GCMs.

In our simulations, the Mediterranean Sea will be warmer and saltier at the end of $21^{st}$ century. This process is gradual but accelerates in the last third of the century. Under the RCP8.5 scenario ROM provides integrated estimates of climate change similar to other models (Table 6). The mean ΔSST projected by ROM under the RCP8.5 scenario is 2.7ºC (ROM_P2-

ROM_P1), close to MPI-ESM simulations, which show an SST increase of 2.8ºC (-LR) and 2.9ºC (-MR). It is also close to the mean increase (2.6ºC) projected by the CMIP5 ensemble of Shaltout and Omstedt (2014) (Table 6). These SST warming estimates also agree with those obtained by Adloff et al. (2015) using a 6-member scenario simulation (3.1ºC warming) and by Darmakari et al. (2019) using a 6-model ensemble (warming from 2.7 to 3.8ºC). In contrast, the ROM_P2 projected mean

SSS change is much smaller than estimated by other authors (Somot et al., 2006; 2008, Adloff et al., 2015; see Table 6). This is related to the dipolar structure of the ΔSSS field (Fig. 13d) pointing to a remarkable salinization in the Eastern Mediterranean, and a slight freshening in a large fraction of the western basin. This is a direct consequence of the North-Atlantic Ocean influence, taken into account through ROM global ocean component, on the thermohaline fields and circulation in the Mediterranean Sea.

The time evolution of the Mediterranean water masses characteristics shows a warming that initially takes place at the surface and gradually penetrates to deeper layers in both eastern and western basins, while there is also a gradual salinity increase, except in the upper 100 m layer of the western basin where there is a freshening. In the Eastern Mediterranean, at depths corresponding to LIW, the warming and salinization accelerate in the last third of the century; this warm and salty signal at intermediate depths subsequently propagates into the western basin. All these changes will have an impact on the

Mediterranean Thermohaline Circulation, which will be addressed in a forthcoming paper.

**5 Conclusions**

In this study, the regional atmosphere-ocean coupled model ROM (Sein et al., 2015) was described and validated for the Mediterranean region. The ROM coupled system has demonstrated benefits compared to other AORCMs without the global ocean. The use of a global ocean model avoids the problems caused by the oceanic boundary conditions, allows a better

understanding of coupling feedbacks between coupled and uncoupled ocean areas (Sein et al., 2015), which is essential for the Mediterranean Sea. For example, the influence of the Modified Atlantic Water in the surface freshening of the Western Mediterranean and the potential impact of the change in properties and production rate of Mediterranean intermediate and deep waters, a mix of which will later exit the Strait of Gibraltar as Mediterranean Outflow, spreading through the North Atlantic and contributing, to a certain extent, to the deep water production in the northern seas. This global ocean approach also provides

an additional "degree of freedom" in the model setup and tuning, which can be helpful, for example, to adjust the ocean component for the better performance within the region of interest. In terms of the climate change projections, the use of a global ocean model could improve AORCMs, which prescribe the global ocean as boundary condition. ROM, as a refined global ocean model coupled with a regional climate change atmospheric model, is able to obtain physically consistent results in the ocean both within and outside of the coupled domain. This prevents the introduction of biases in the results that are

typical of regional ocean models, which implement lateral boundary conditions provided by coarser global AOGCMs scenario simulations (Sein et al. 2015).

The experiment in which our model is driven by ERA-Interim shows good performance in simulating the present climate. ROM is able to reproduce the main characteristics of the Mediterranean Sea, providing a physically consistent estimation of the average behavior, seasonal cycle and interannual variability of both atmospheric and oceanic parameters. However, there is place for further improvement in reducing certain biases (SST and MSLP) by isolating the causes through targeted sensitivity experiments. At this point, we have found that an appropriate modification of the optical properties of the water leads to a reduction of SST bias. For instance, the inclusion of a marine biogeochemistry model (i.e. HAMOCC) improves the ROM_P0 SST performance.

The model simulates explicitly the exchange of water through the Gibraltar and Dardanelles Straits taking into account the signals from the neighboring basins (Atlantic Ocean and Black Sea), which is essential to include the large-scale feedbacks in the climate signal of the Mediterranean. Moreover, ROM shows improvements in reproducing local and mesoscale features in the Mediterranean Sea in contrast to ESMs.

Our analysis of the simulations driven by the MPI-ESM RCP8.5 scenarios shows that by the end of the 21$^{st}$ century the Mediterranean Sea will be warmer and saltier throughout most of the basin. The temperature in the upper ocean layer during the period 2070-2099 will increase by 2.7ºC in comparison with the 1976-2005 control period, while the mean salinity will increase by 0.2 psu. The warming that initially takes place at the surface propagates gradually to the deeper layers. Furthermore, it is very remarkable that the Western Mediterranean surface layer presents a decreasing salinity tendency, opposite to the rest of the Mediterranean. There is a change in the LIW characteristics, which propagates from the Eastern Mediterranean to the west, pointing to MTHC changes in the future.

An important disadvantage of the proposed model is that the biases and internal variability generated in the global domain can influence the results in the coupled domain, making it difficult to separate the source of bias.

Finally, we conclude that the ROM is a powerful model system that can be used to estimate possible impacts of climate change on regional scale. In the future, we plan to use our ROM coupled system to characterize and analyze the climate variability of deep water formation in the Mediterranean Sea.

**Acknowledgements**

Simulations were done at the German Climate Computing Centre (DKRZ). DS was supported by PRIMAVERA funding received from the European Commission under Grant Agreement 641727 of the Horizon 2020 research program and by the state assignment of FASO Russia (theme No. 0149-2019-0015). William Cabos has been funded by the Spanish Ministry of Science, Innovation and Universities, the Spanish State Research Agency and the European Regional Development Fund, through grant CGL2017-89583-R.

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

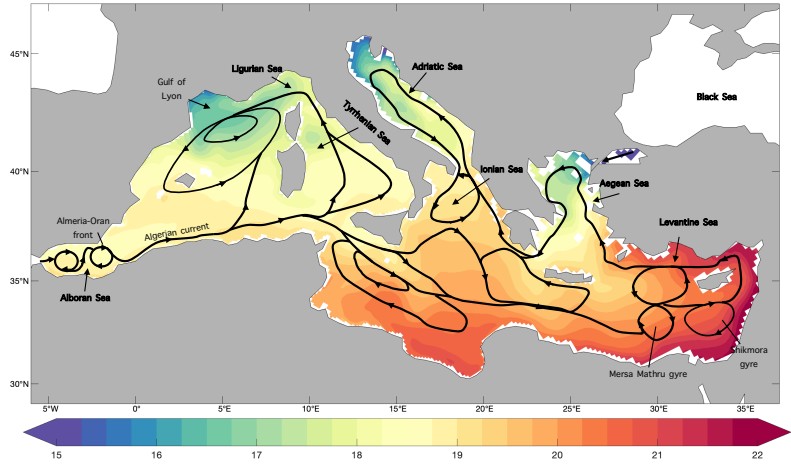

**Figure 1: Mediterranean basin: 1980-2012 mean SST (ºC) and upper ocean currents (Based on Tomczak and Godfrey, 1994).**

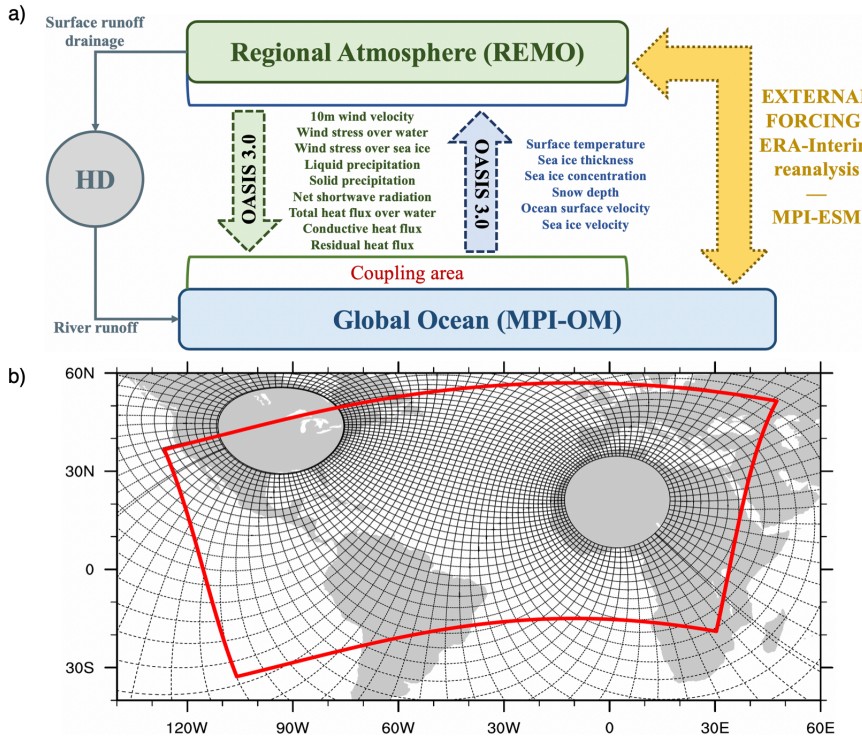

**Figure 2: (a) ROM coupling scheme. (b) Atmospheric and oceanic ROM grids. MPI-OM variable resolution grid (black lines, drawn every twelfth), REMO domain (red line).**

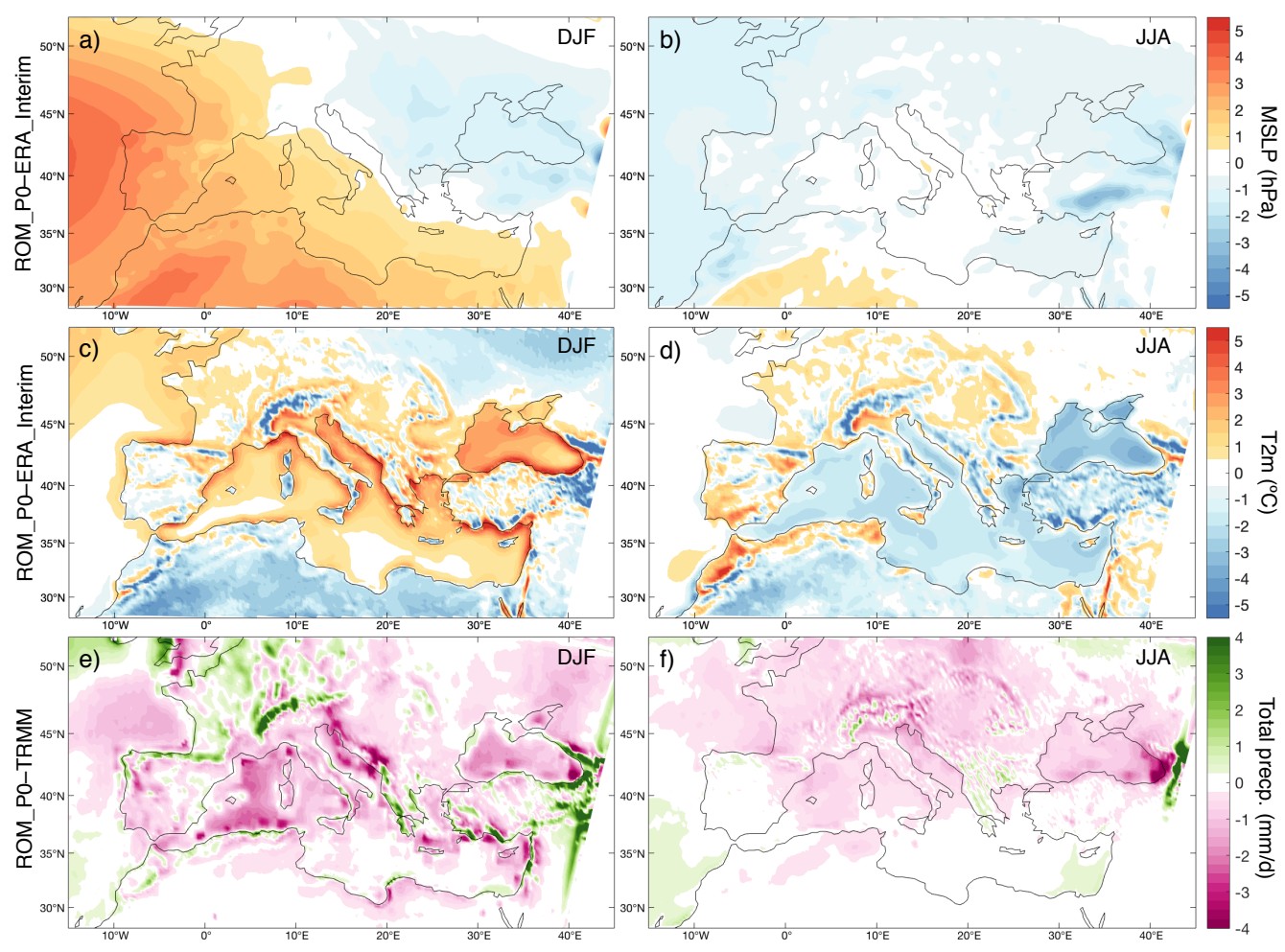

**Figure 3: Differences between ROM_P0-ERA-Interim and TRMM for the 1980-2012 period in mean sea level pressure (MSLP, hPa) (upper row), near-surface (2m) temperature (T2m, ºC) (middle) and precipitation (mm/d) (bottom). Left, DJF; right, JJA.**

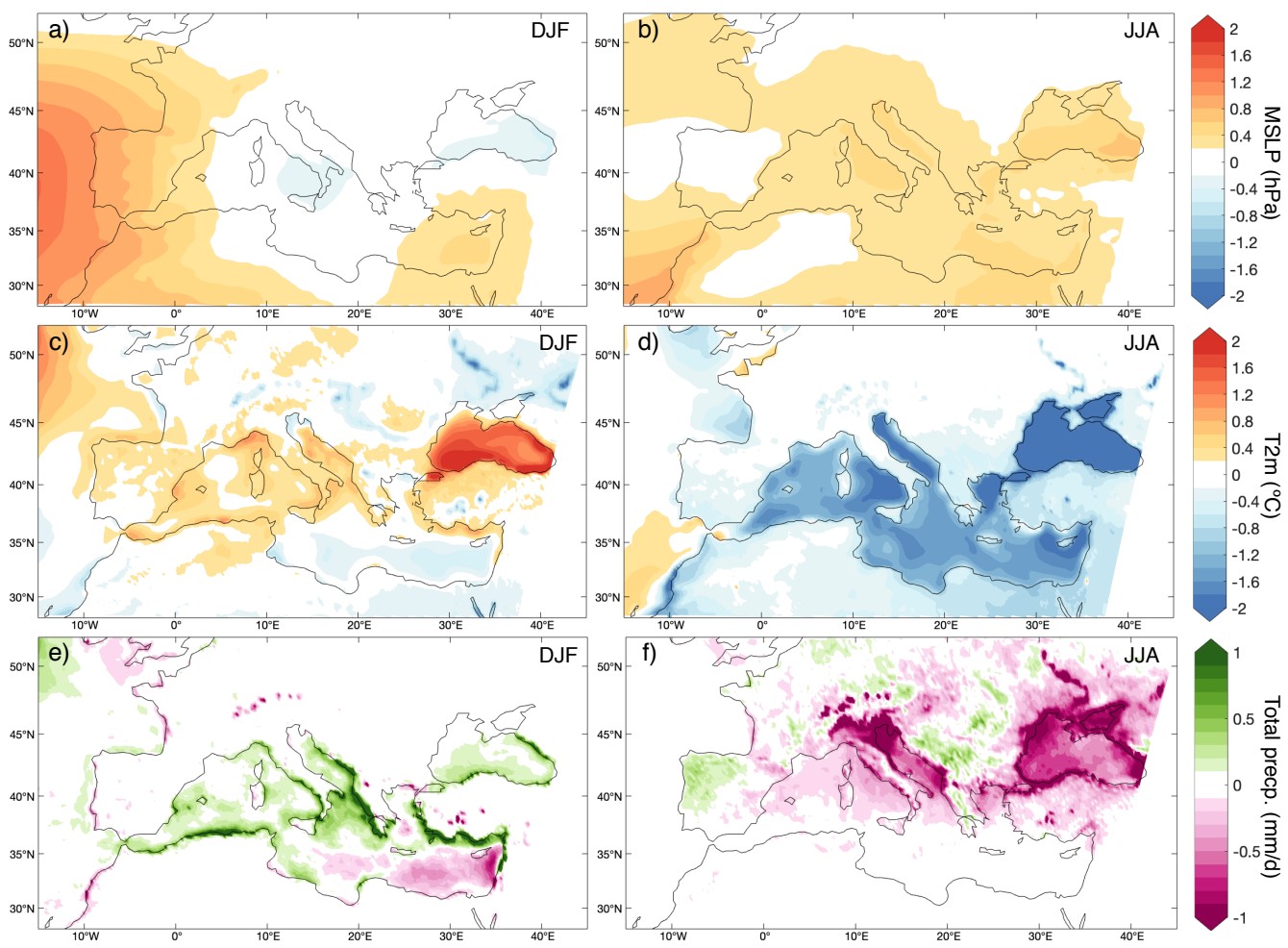

Figure 4: Differences between ROM_P0 and stand-alone REMO forced by ERA-Interim for the 1980-2012 period in mean sea level pressure (MSLP, hPa) (upper row), near-surface (2m) temperature (T2m, ºC) (middle) and precipitation (mm/d) (bottom). Left, DJF; right, JJA.

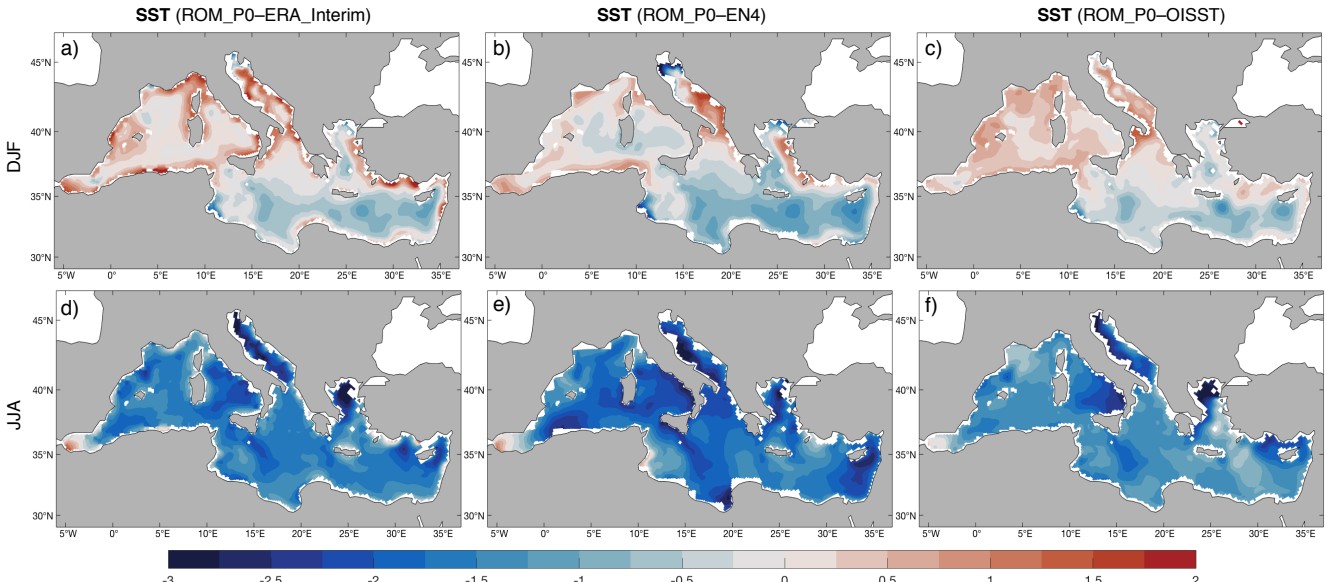

**Figure 5: Difference between the ROM_P0 SST (ºC) and the different climatologies (ERA-Interim [left], EN4 [middle] and OISST [right]) in winter (DJF, top), and summer (JJA, bottom).**

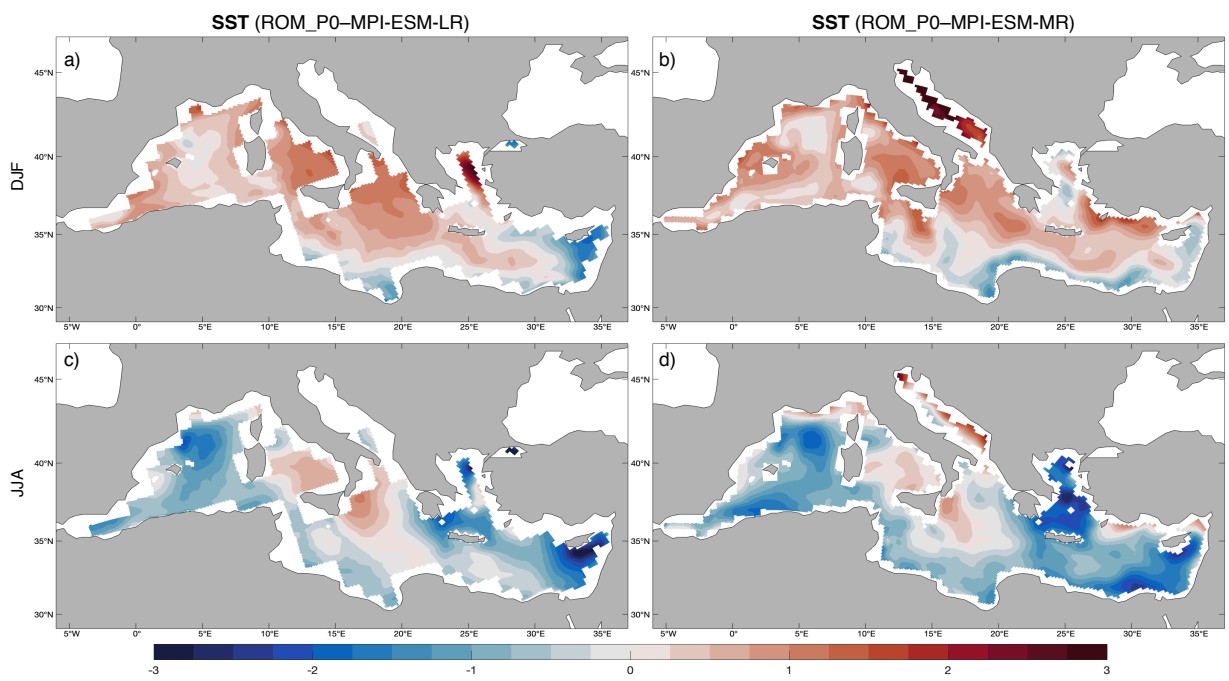

5 **Figure 6: SST difference (ºC) between ROM_P0 and MPI-ESM-LR (left) and -MR (right) in winter (DJF, top), and summer (JJA, bottom).**

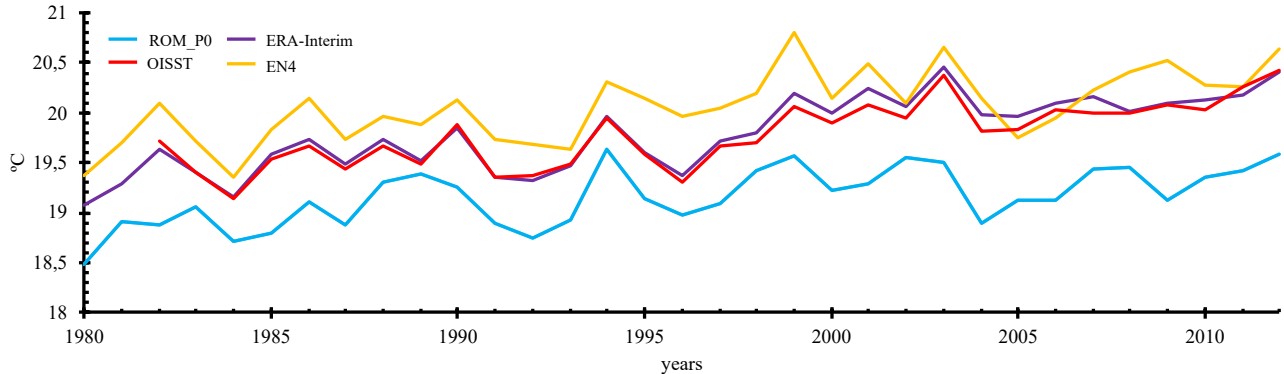

**Figure 7: Time series of yearly mean (1980-2012) SST (ºC) averaged over the Mediterranean basin. ROM_P0 (blue), OISST (red), ERA-Interim (purple) and EN4 (yellow).**

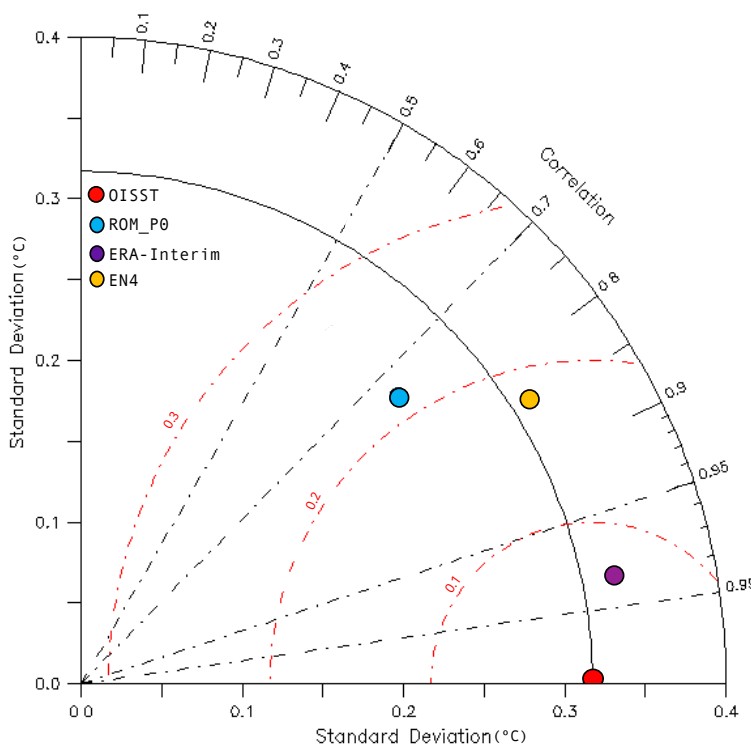

**Figure 8: Taylor Diagram for Mediterranean SST during the 1982-2012 period. The diagram summarizes the relationship between standard deviation (ºC), correlation (r) and RMSE (red lines) (ºC) for all datasets. The gridded OISST was employed as reference.**

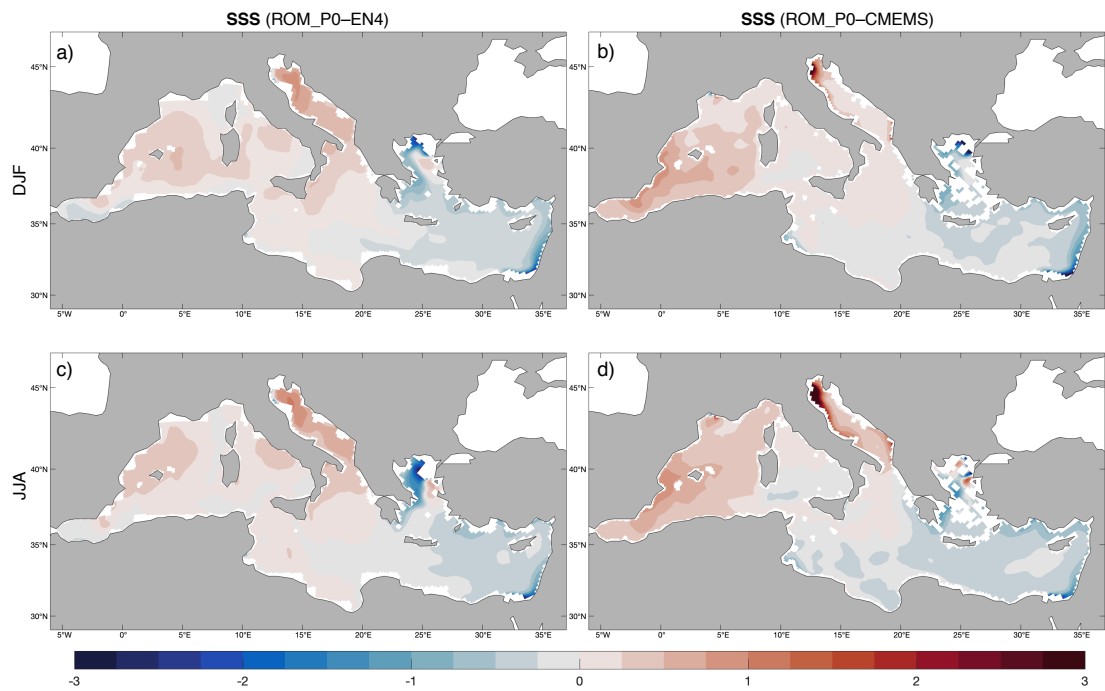

**Figure 9: SSS (psu) difference between the ROM_P0 climatologies (EN4 [left] and CMEMS [right]) in winter (DJF, top), and summer (JJA, bottom).**

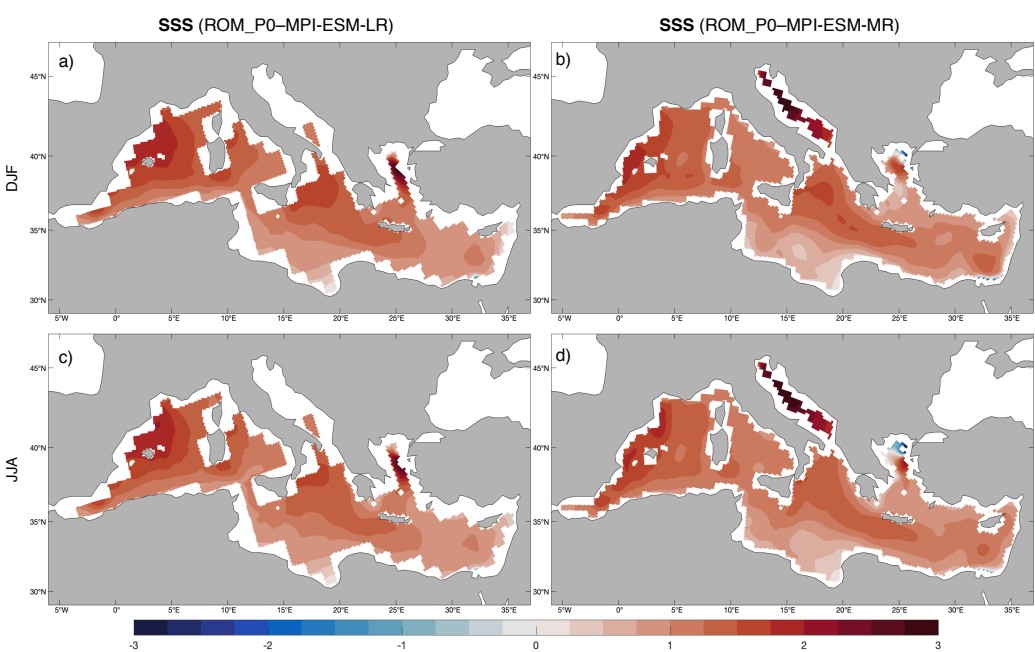

5   **Figure 10: SSS (psu) difference between the ROM_P0 and MPI-ESM-LR (left), -MR (right) in winter (DJF, top), and summer (JJA, bottom).**

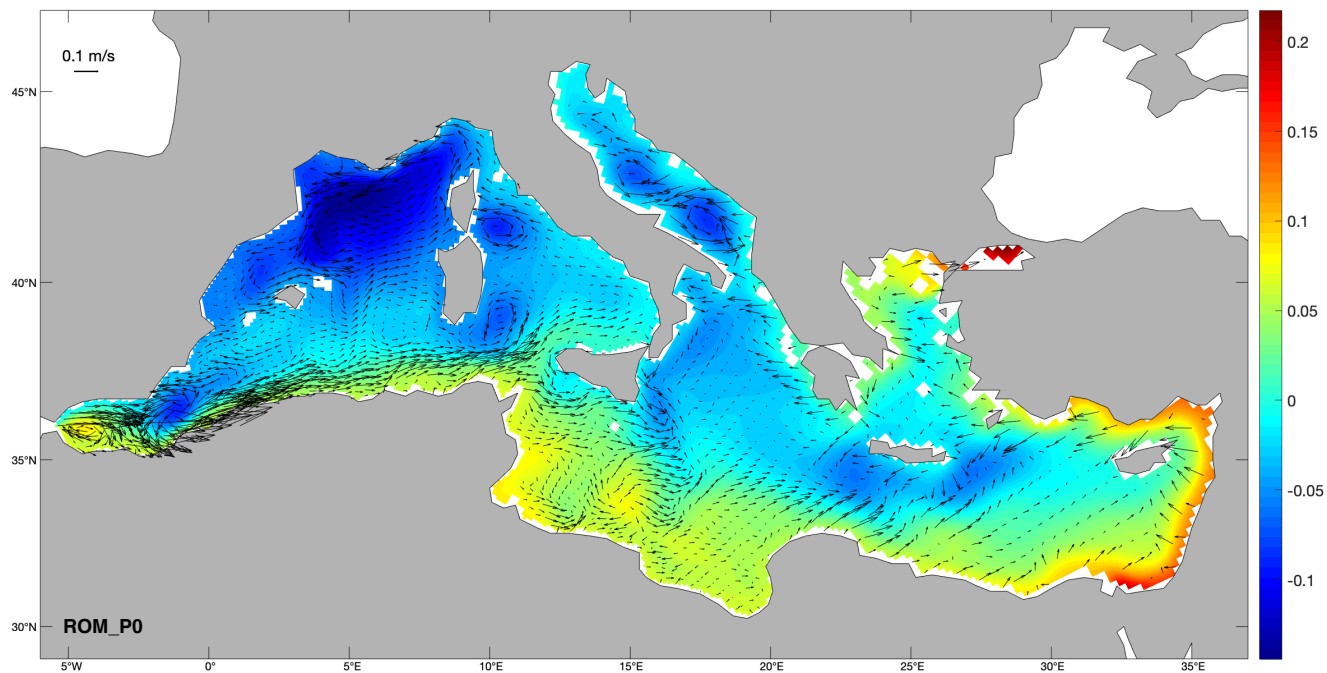

**Figure 11: Mean (1980-2012) ROM_P0 SSH (m) and horizontal current velocity at 31 m depth (vectors, in m/s). Only every sixth vector is plotted.**

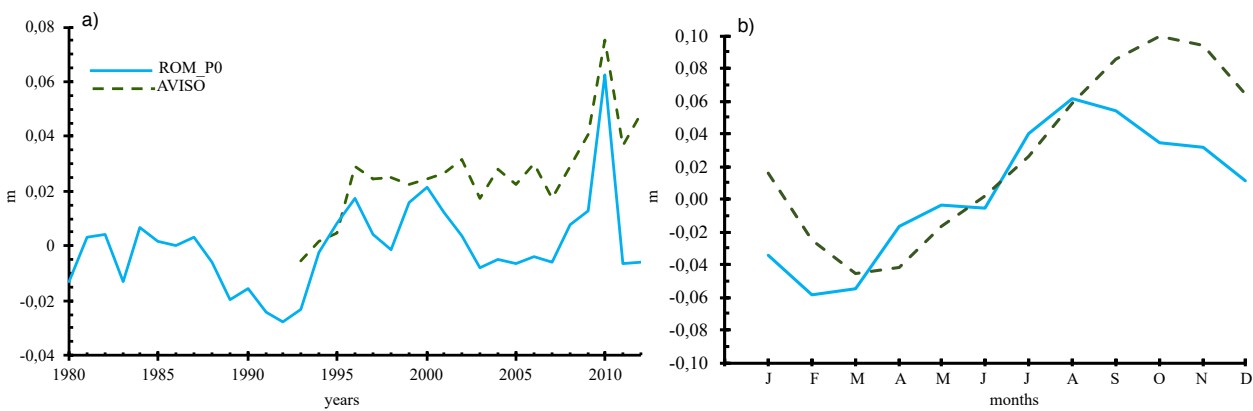

**Figure 12: Time series of mean (1980-2012) sea-level anomalies averaged over the Mediterranean basin (left, in m). For ROM_P0 (blue), the dynamic SSH is added to the thermosteric term. Model data are compared to observations (AVISO, green dashed). ROM_P0 seasonal cycle is compared to AVISO data (right).**

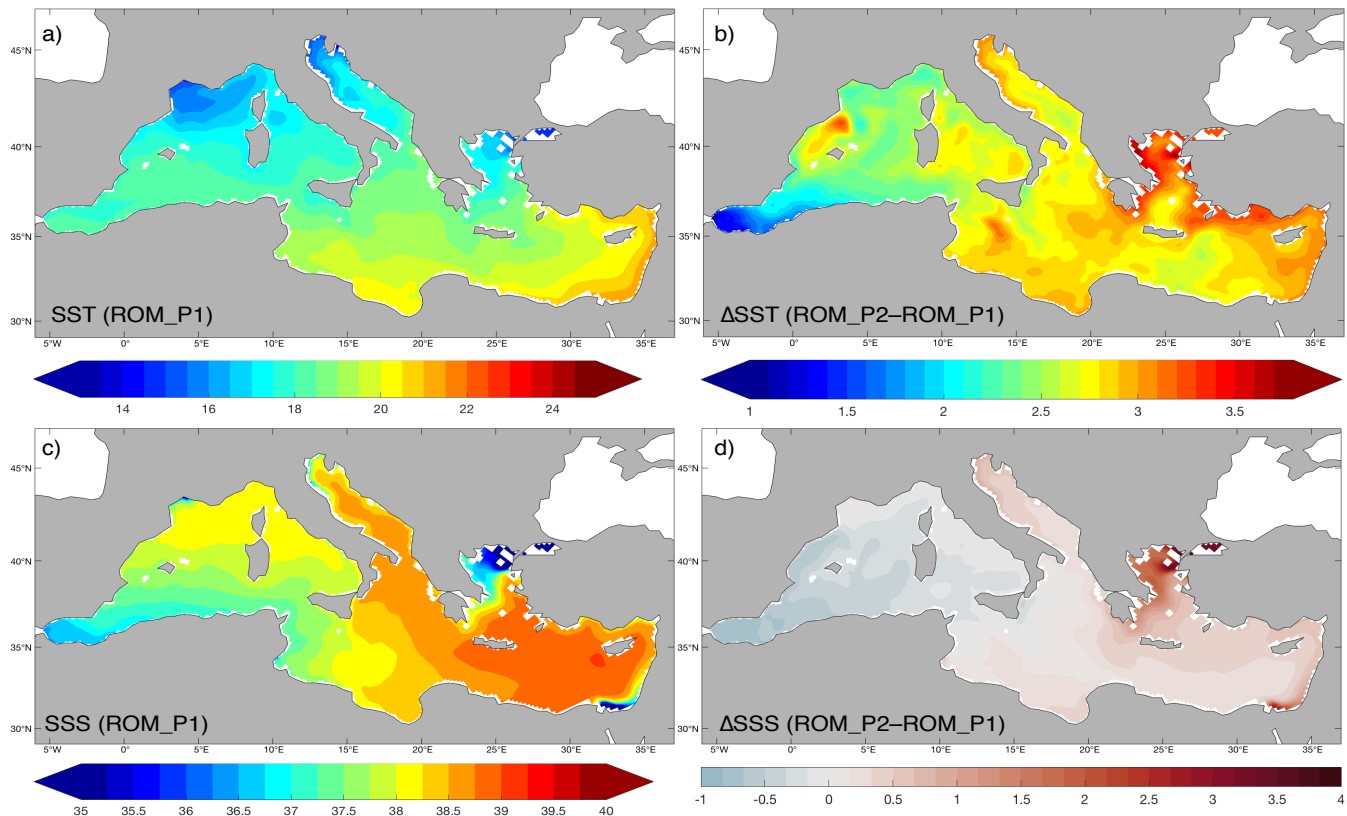

**Figure 13: Mean SST (in °C, top left) and SSS (in psu, bottom left), averaged over the 1976-2005 period (ROM_P1). Difference between mean SST (in °C, top right) and SSS (in psu, bottom right) RCP8.5 projection (2070-2099, ROM_P2) and present climate (1976-2005, ROM_P1).**

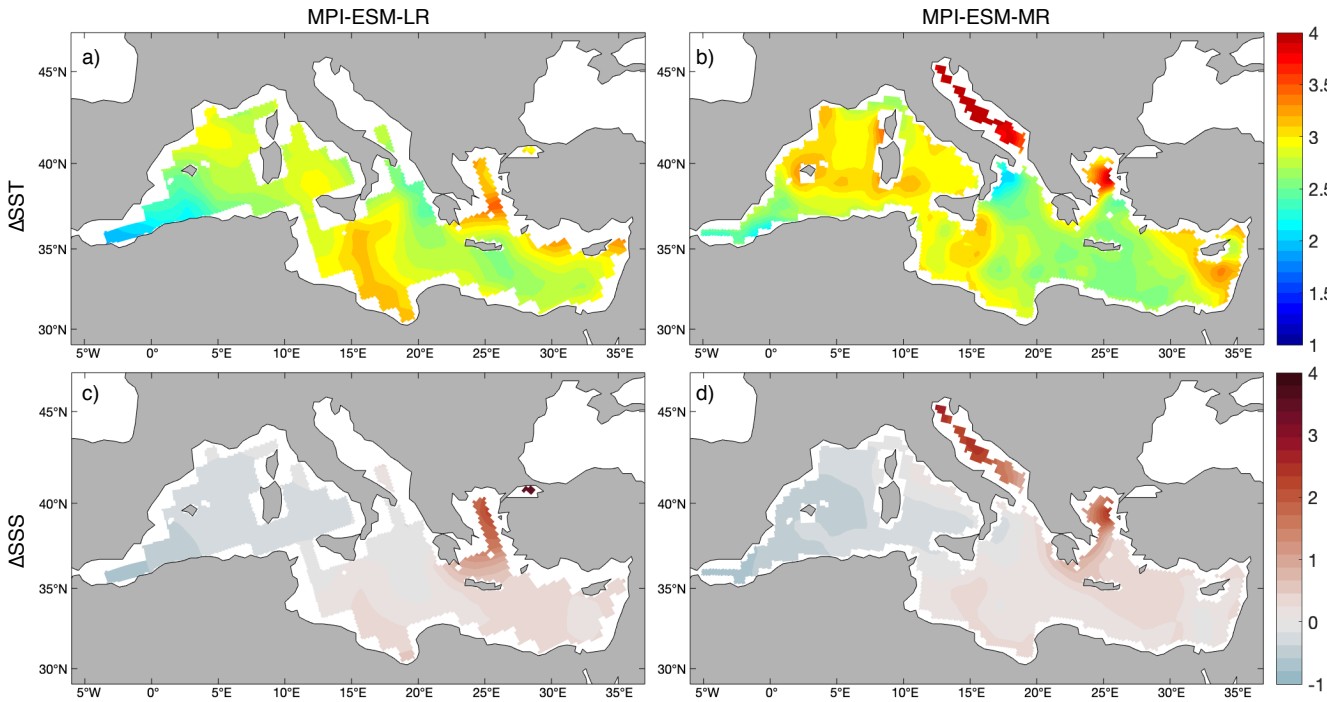

**Figure 14: SST (in ºC, upper row) and SSS (in psu, bottom) MPI-ESM-LR (left) and -MR (right) anomaly fields estimated as the difference between the average of the RCP8.5 projection (2070-2099) and present climate (1976-2005).**

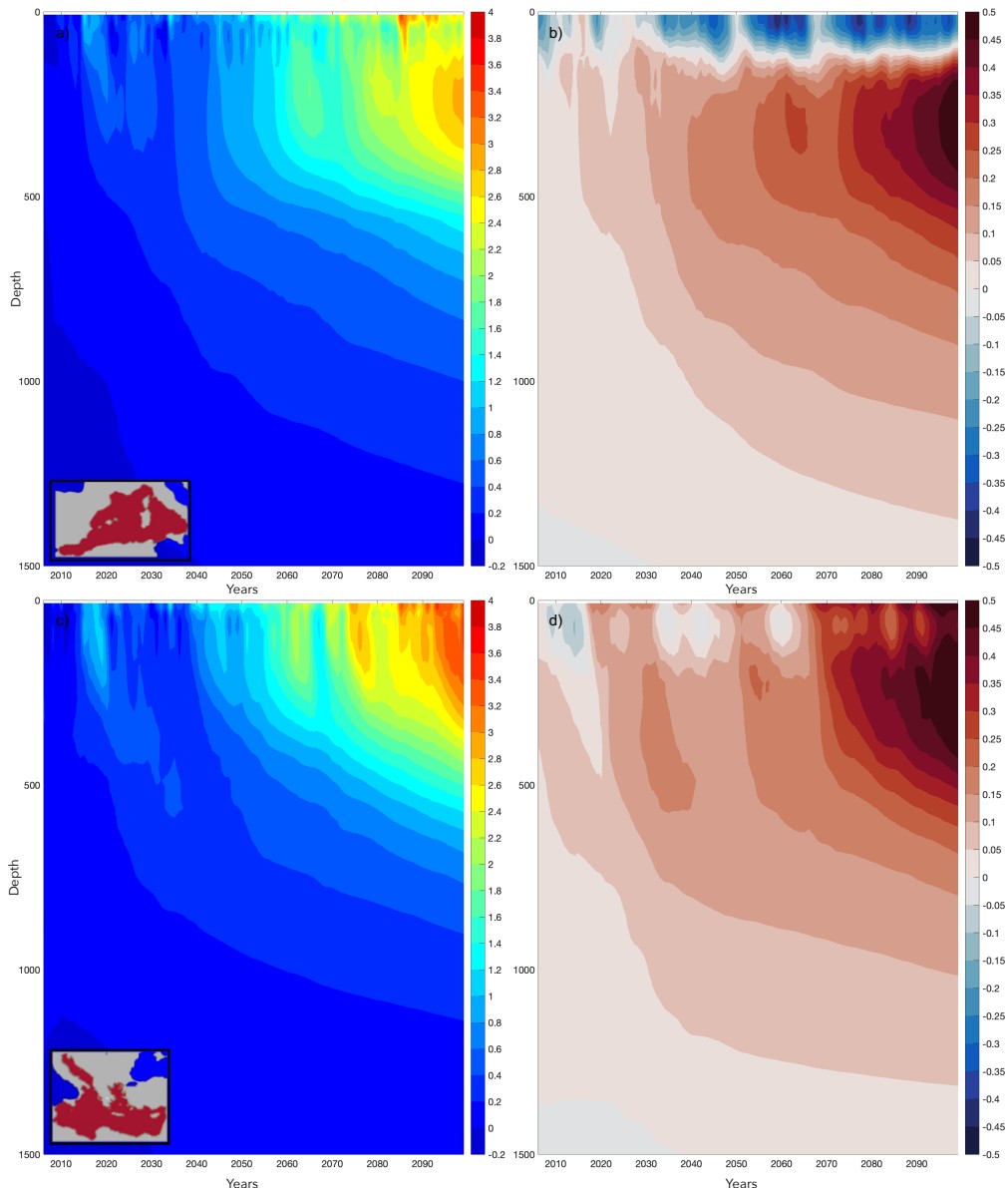

**Figure 15: Temporal evolution of mean temperature (in ºC, left) and salinity (in psu, right) throughout the twenty-first century at Western (upper row) and Eastern Mediterranean (bottom).**

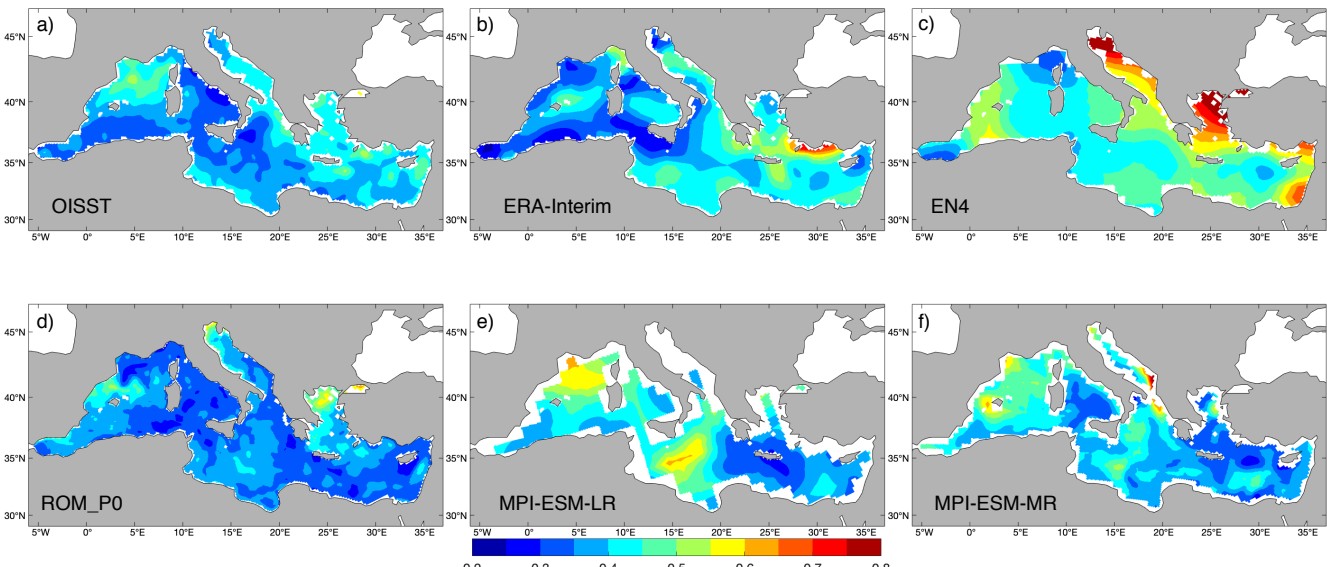

**Figure 16: Yearly mean SST standard deviation (in ºC) for 1982-2012 period: OISST (a), ERA-Interim (b), EN4 (c), ROM_P0 (d), MPI-ESM-LR (e) and MPI-ESM-MR (f).**

**Table 1.** Characteristics of ROM atmosphere-ocean regional coupled model used in this study. Modified from Darmaraki et al. (2019). For details see Sein et al. (2015).

| | |
|---|---|
| **Institute** | AWI/GERICS |
| **Driving GCM** | MPI-ESM-LR |
| **Med. Sea Model** | MPI-OM |
| **Ocean Res.** | 7-25 km |
| **Num. of z-levels (ocean)** | 40 |
| **SST (1st layer depth)** | 16 m |
| **Timestep (ocean)** | 900 s |
| **Atmosphere model** | REMO |
| **Atmosphere Res.** | 25 km |
| **Timestep (atmosphere)** | 120 s |
| **Coupling frequency** | 60 min |

**Table 2. Datasets used in the ROM validation.**

| | **Parameters** | **Period** | **Spatial resolution** | **Datasets** |
|---|---|---|---|---|
| **Atmosphere** | MSLP | 1980-2012 | 80 km (T255 spectral) | ERA-Interim (Dee et al., 2011) |
| | T2m | 1980-2012 | 80 km (T255 spectral) | ERA-Interim (Dee et al., 2011) |
| | Precipitation | 1997-2012 | 1/4º x 1/4º | TRMM (Huffman et al., 2014) |
| **Ocean** | SST | 1982-2012 | 1/4º x 1/4º | OISST (Reynolds et al., 2007) |
| | | 1980-2012 | 80 km (T255 spectral) | ERA-Interim (Dee et al., 2011) |
| | | 1980-2012 | 1º x 1º | EN4 v.4.1.1 (Good et al., 2003; Gouretski and Reseghetti, 2010) |
| | | 1980-2012 | 1.5º x 1.5º / 0.4º x 0.4º | MPI-ESM-LR and -MR (Giorgetta et al., 2013) |
| | SSS | 1980-2012 | 1º x 1º | EN4 v.4.1.1 (Good et al., 2003; Gouretski and Reseghetti, 2010) |
| | | 1980-2012 | 1/16º x 1/16º | CMEMS (Fratianni et al., 2015) |
| | | 1980-2012 | 1.5º x 1.5º / 0.4º x 0.4º | MPI-ESM-LR and -MR (Giorgetta et al., 2013) |
| | SSH | 1993-2012 | 1/4º x 1/4º | SSALTO/DUACS L4 |

**Table 3. Trend computed from yearly means during 1980-2012 by the different analysis into the Mediterranean Sea.**

| | ROM_P0 | OISST | ERA Interim | EN4 | MPI-ESM-LR | MPI-ESM-MR |
|---|---|---|---|---|---|---|
| **ºC/year** | +0.016 | +0.027 | +0.029 | +0.022 | +0.028 | +0.020 |

**Table 4. Water balance and exchange flows for the Mediterranean Sea according to ROM_P0, RCSM4 and observation-based estimates. All results are presented in Sverdrups (Sv).**

| Parameters | 1980-2012 mean ROM_P0 | RCSM4 Sevault et al., (2014) | Estimates |
|---|---|---|---|
| Evaporation | 0.093 | 0.110 | 0.086-0.089 (Sánchez-Gómez et al., 2011) |
| Precipitation | 0.034 | 0.040 | 0.020-0.047 (ibid) |
| Runoff | 0.006 | 0.010 | - |
| E-P | 0.059 | 0.070 | 0.039-0.069 (ibid) |
| E-(P+R) | 0.053 | 0.060 | - |
| Gibraltar in | 0.554 | 0.850 | 0.81 (Soto-Navarro et al., 2014) |
| Gibraltar out | 0.524 | 0.800 | 0.78 (ibid) |
| Gibraltar net | 0.030 | 0.050 | 0.04-0.10 (ibid) |
| Dardanelles in | 0.132 | - | - |
| Dardanelles out | 0.109 | - | - |
| Dardanelles net | 0.023 | 0.007 | 0.008-0.01 (Sánchez-Gómez et al., 2011) |

**Table 5. Resolution of the different models used in this study to discuss ROM.**

| Model | Model configuration | Atmosphere/Ocean Resolution | References |
|---|---|---|---|
| MGME ensemble | Global | 1 - 4º / - | Giorgi and Lionello, (2018) |
| PROTHEUS | AORCM | 30 km / 13 km | Artale et al. (2010) |
| LMDz-NEMO-Med | AORCM | 30 km / 9 – 12 km | L'Hévéder et al. (2013) |
| WRF RCM | RCM | 50 km / - | Di Luca et al. (2014) |
| CNRM-RCSM4 | AORCM | 50 km / 9 – 12 km | Sevault et al. (2014) |
| RCM11 | RCM | 12 km / - | Fantini et al. (2016) |
| RCM44 | RCM | 50 km / - | Fantini et al. (2016) |

**Table 6. Mediterranean Sea spatial averaged changes in SST and SSS at the end of the twenty-first century as compared with the present climate**.

| | Scenario | ΔSST (ºC) | ΔSSS (psu) |
|---|---|---|---|
| **ROM** | RCP8.5 | +2.7 | +0.2 |
| **MPI-ESM-LR** | RCP8.5 | +2.8 | +0.1 |
| **MPI-ESM-MR** | RCP8.5 | +2.9 | +0.1 |
| **Thorpe and Bigg (2000)** | $2XCO_2$ | +4 | - |
| **Somot et al. (2006)** | A2 | +2.50 | +0.33 |
| **Somot et al. (2008)** | A2 | +2.60 | +0.43 |
| **Shaltout and Omstedt (2014)** | RCP2.6 | +0.5 | - |
| **(ibid)** | RCP4.5 | +1.15 | - |
| **(ibid)** | RCP6.0 | +1.42 | - |
| **(ibid)** | RCP8.5 | +2.6 | - |
| **Adloff et al. (2015)** | A2 | +2.53 | +0.48 |
| **(ibid)** | A2-F | +2.97 | +0.69 |
| **(ibid)** | A2-ARF | +2.97 | +0.89 |
| **(ibid)** | B1-ARF | +1.73 | +0.70 |
| **Darmaraki et al. (2019)** | RCP8.5 | +3.1 | - |