# Peer review of "The climate change signal in the Mediterranean Sea in a regionally coupled atmosphere-ocean model"

_Ocean Science, 2019_

## Referee Comment (RC1) · Anonymous Referee #1 · 11 Jun 2019

The manuscript evaluates the set-up of the regional climate system model ROM, i.e. a coupled atmosphere-land-ocean-river climate model on the basis of the RCM REMO, focusing over the Mediterranean Sea. The system is also applied in a future projection driven by the global MPI-ESM using the scenario RCP8.5. The evaluation is important to document, and the projection is worth to be discussed. The manuscript is also well written and structured. Still, I miss some scientific content. For example, it is promised (in the abstract, 1st sentence) that the role of the ocean feedbacks to the atmosphere is assessed. It would be of interest to learn about that and what impact the interactive ocean might have on the REMO's atmosphere simulation. But this discussion is missing. Only one projection run without any reference (e.g. ocean driven offline by

stand-alone REMO in the coupling area) or discussion with respect to published results is a bit weak. Also, missing is the discussion with respect to newer literature (e.g. Damarski et al. 2019 with common co-authorship and use of ROM, too). There are no recent references listed and discussed. Especially, literature from the Med-CORDEX activity (which is mentioned) special issue (see Somot et al. 2018) is ignored at all. I conclude the manuscript is in need of a full update and thorough revision.

Darmaraki, Sofia, Samuel Somot, Florence Sevault, Pierre Nabat, William David Cabos Narvaez, Leone Cavicchia, Vladimir Djurdjevic, Laurent Li, Gianmaria Sannino, and Dmitry V Sein. (2019). Future Evolution of Marine Heatwaves in the Mediterranean Sea." Climate Dynamics 45 (9–10), 1–22. doi:10.1007/s00382-019-04661-z.

Somot S., P. Ruti, B. Ahrens, E. Coppola, G. Jordà, G. Sannino, F. Solmon (2018). Editorial for the Med-CORDEX special issue. Clim. Dyn., 51: 771. DOI: 10.1007/s00382-018-4325-x

---

## Referee Comment (RC2) · Anonymous Referee #2 · 20 Jun 2019

Comments for the manuscript **"The climate change signal in the Mediterranean Sea in a regionally coupled ocean-atmosphere model"**

**General comments**

This study has two main parts, (1) a performance evaluation for the regional coupled model REMO-OASIS-MPI-OM (ROM) which has been set up for the Mediterranean Sea and forced by the ERA-Interim reanalysis data for the present climate, and (2) a production of climate change signals in sea water temperature and salinity using ROM forced by the global model MPI-ESM with the RCP8.5 scenario. Main results are (1) simulations of ROM forced by ERA-Interim show a good representation of the present Mediterranean Sea climate and (2) climate change signal in the Mediterranean Sea is found to be similar to previous studies.

New findings of the study are not highlighted, neither in the abstract nor in the main text. In the Introduction, the authors mentioned that using the global ocean model could help to avoid some problems associated with the open boundary conditions for the Mediterranean Sea. However, no reference of previous studies was cited and no results were shown to support the argument. Moreover, this study compared the climate change signal produced by ROM with previous studies. The authors concluded that the ROM simulations under RCP 8.5 scenario provided integrated estimates of climate change similar to other models. The study would be more highlighted if the authors recommend ROM as one more ensemble model member to provide more robust information of climate change signal for the Mediterranean Sea.

I suggest accepting the paper for publication after major revisions are made.

**Major remarks**

- It's not clearly described how the simulation (ROM forced by ERA-Interim for 1982-2012) and the hindcast (ROM forced by MPI-ESM for 1976-2005) were set up and used in the study. It's confusing to the reader that authors seem to have calculated the change signal by substracting the climate projection for 2070-2099 to the climate simulation (ROM forced by the reanalysis data ERA-Interim) for 1976-2005 (which is an invalid calculation of the climate change signal). Please clarify! A suggestion could be: The present simulation of ROM forced by the reanalysis data ERA-Interim should be named as ROM_P0, and the hindcast of ROM force by MPI-ESM for the present climate as ROM_P1, the climate projection for 2070-2099 as ROM_P2. Consequently, the climate change signal should be yielded by subtracting ROM_P1 from ROM_P2.

- Through the Results section, details of bias, difference and changes of ROM compared with other data sets were shown which could help readers to have a good overview of ROM performance. However, there is a lack of deep analysis about potential reasons for such biases or differences. For example, on Page 9 Line 23-24: "MPI-ESM-LR and ROM show a similar distribution …" Can the authors speculate why? In previous parts, usually MPI-ESM-MR and ROM are similar. Or what is the potential reason for the different trends in Western and Eastern Mediterranean Sea mentioned on Page 10 Line 8-10?

- The paragraph on Page 7 Line 6-10 only describes how the SST seasonal cycle amplitude of ROM is different to (i.e. smaller than) MPI-ESM but it doesn't support the statement of the authors that ROM is better than MPI-ESM due to the higher resolution. Moreover, "ROM *overestimates* the SST simulated be MPI-ESM" should be rephrased because "overestimates" is often used while comparing with an observation. Please rewrite this part.

-  A comparison between SST time series of MPI-ESM with OISST (Figure 6) or an analysis about temporal correlation between MPI-ESM and OISST (Figure 7 & 15) make no sense as MPI-ESM doesn't know anything about SST of a certain 'real' year. Please remove these figures or at least the part of MPI-ESM and focus more on other results.

- Section 2: details of basis configuration of REMO, MPI-OM such as horizontal and vertical resolution as well as running time step should be described. It's also necessary to give a list of variables exchanged between REMO and MPI-OM via OASIS.

- Introduction: Page 2 Line 10-27 simply listed the previous study without any results mentioned. I suggest to summarize this part of introduction and mention more details about only the previous studies which gave information of climate change signal in SST and SSS that relates to the main topic of the current study. The authors should make it clearer what has been done in the past, what is still missing and why this study is important.

**Minor Comments**

- Should it be "AORCMs" for coupled "atmosphere-ocean regional climate models" as used in many previous studies instead of RAOCMs?

- Page 1 Line 24 is repeated at line 27.

- Page 2 Line 34 & page 3 Line 4 have the same typo of "Asses"

- Please follow the citation rule of the journal. At several places, "el al., (20xx)" was used where either "," or "(" is needed.

- Page 2 Line 22: "Finally, Sevault et al. …": does it mean it's the last fully coupled regional climate system model has been developed?

- Page 2 – Line 30 – Abbreviation MPI-OM is only introduced on page 3, Sect. 2. Please remove here. The abbreviation ROM has to be described as it is mentioned the first time in the main text (the abstract does not count).

- Sequence of 2.1 and 2.2 should be switched as REMO was mentioned before MPI-OM at the beginning of Section 2 Methods and also in sequence of abbreviation "AORCM". The author should also think about the sequence of "ocean-atmosphere" or "atmosphere-ocean" in the title to ensure the consistence for the whole manuscript. In addition, model abbreviations and references are provided in Sect. 2, and, hence can be removed from Sect. 2.1 and 2.2.

- Page 3 Line 9: For this work, the ROM climate model (Sein et al. 2015) has been used.

- Page 3 Line 12 - …, the soil model of REMO (Rechid …

- Page 3 Line 13: Which version of OASIS was used?

- Page 3 Line 30: "REMO's prognostic variables are …": for what are they important to be mentioned here? More important should be which variables are exchanged between REMO and MPI-OM via OASIS.

- Page 4 Line 14: double "with"

- Page 4 Line 16: … with a resolution of about 25 km …

- Page 4 Line 18: why different coupling time steps (3 hrs & 24 hrs) are using?

- Page 4 Line 21: "… scenario were analysed."

- Page 5 Line 18: …of ROM's potential to improve the …

- Page 6 Line 8: "can be found *in* DJF"?

- Page 6 Line 21: deviation of 3.5 mm/d is not a small amount. This corresponds to ~315 mm/season. Please comment more thoroughly!

- Page 7 Line 3: … than expected …

- Page 7 Line 5: … have been done…

- Page 8 Line 10: "It is clearly *seen* how …"

- Page 8 Line 11: "penetrate to the Western Mediterranean by the African continent": what do you mean?

- Page 8 Line 14-15: sentence is incomplete.

- Figure 2: HD is missing. How are u & v surface currents passed from MPI-OM used in REMO?

- Figure 6, 7 & 15: please remove the MPI-ESM_LR and MPI-ESM_MR as their SST temporal timeseries have non-sense.

- Figure 10: vector is too small. Why level 31m depth was chosen to be shown here?

- Figure 12: why do not show figures for DJF and JJA separately? A strong SST bias in summer of ROM (Fig.4) may affect the trend analysis if it's not system bias.

---

## Author Comment (AC3) · 25 Jul 2019

Please find attached the *pdf with the revised manuscript marked-up and a supplementary figure.

Please also note the supplement to this comment:
https://www.ocean-sci-discuss.net/os-2019-42/os-2019-42-AC3-supplement.pdf
* * *
[Figure]

[Figure]

**Supplementary figure: Mean SST (in °C) in winter (DJF, top left) and summer (JJA, bottom left) averaged over the 1976-2005 period. Difference between mean SST (in °C) RCP8.5 projection (2070-2099, ROM_P2) and present climate (1976-2005, ROM_P1) for DJF (top right) and JJA (bottom right).**

**Fig. 1.** Supplementary figure

---

## Author Response (AR1)

My coauthors and I would like to thank the reviewer for the advises that greatly have helped to improve our manuscript "The climate change signal in the Mediterranean Sea in a regionally coupled atmosphere-ocean model". The review comments point-by-point response are reported in blue whereas our answers are in bold letters.

**Reviewer's comments:**

**Referee #1**: «…it is promised (in the abstract, 1st sentence) that the role of the ocean feedbacks to the atmosphere is assessed. It would be of interest to learn about that and what impact the interactive ocean might have on the REMO's atmosphere simulation. But this discussion is missing. Only one projection run without any reference (e.g. ocean driven offline by stand-alone REMO in the coupling area) or discussion with respect to published results is a bit weak. »

**Response**: Thank you for the suggestion. Initially, we only tried to assess the downscaled climate change signal in the ocean and not in the atmosphere, it could be wrongly expressed in the abstract "Line 1". In the section 3.1 of the revised manuscript we have added the differences between ROM (coupled) and stand-alone REMO forced by ERA-Interim (see Fig. 4) with the aim to analyze the role of the interactive ocean on the REMO's atmosphere simulation. In addition, you can find the discussion of our new results later in section 4.

**Referee #1**: «Also, missing is the discussion with respect to newer literature (e.g. Darmaraki et al. 2019 with common co-authorship and use of ROM, too). There are no recent references listed and discussed. Especially, literature from the Med-CORDEX activity (which is mentioned) special issue (see Somot et al. 2018) is ignored at all. »

**Response**: Thank you. We have updated the introduction and the discussion in the revised manuscript. We discuss the relevant literature (e.g. Akhtar et al. 2018; Macias et al. 2018; Darmakari et al. 2019) from the recent Med-CORDEX activity special issue to provide a more critical vision respect to the state of the art (see section 4).
* * *
**Major remarks**

**Referee #2**: «It's not clearly described how the simulation (ROM forced by ERA-Interim for 1982-2012) and the hindcast (ROM forced by MPI-ESM for 1976-2005) were set up and used in the study. It's confusing to the reader that authors seem to have calculated the change signal by subtracting the climate projection for 2070-2099 to the climate simulation (ROM forced by the reanalysis data ERA-Interim) for 1976-2005 (which is an invalid calculation of the climate change signal). Please clarify! A suggestion could be: The present simulation of ROM forced by the reanalysis data ERA-Interim should be named as ROM_P0, and the hindcast of ROM force by MPI-ESM for the present climate as ROM_P1, the climate projection for 2070-2099 as ROM_P2. Consequently, the climate change signal should be yielded by subtracting ROM_P1 from ROM_P2. »

**Response**: Thank you for the suggestion. We have added names for each of our simulations in the revised manuscript (Page 4 Line 24-30):

ROM_P0 → ROM forced by ERA-Interim (1980-2012).
ROM_P1 → historical forced by MPI-ESM-LR (1976-2005).
ROM_P2 → future projection forced by RCP 8.5 (2070-2099).

The climate change signal was calculated as the differences between (ROM_P2 – ROM_P1).

**Referee #2**: «Through the Results section, details of bias, difference and changes of ROM compared with other data sets were shown which could help readers to have a good overview of ROM performance. However, there is a lack of deep analysis about potential reasons for such biases or differences. For example, on Page 9 Line 23-24: "MPI-ESM-LR and ROM show a similar distribution ..." Can the authors speculate why? In previous parts, usually MPI-ESM-MR and ROM are similar. Or what is the potential

reason for the different trends in Western and Eastern Mediterranean Sea mentioned on Page 10 Line 8-10? »

**Response**: As we mention in the paper (section 2.3) the ROM_P1 was forced by MPI-ESM-LR, that is the main reason why it shown similar distributions (see Page 13 Line 25-27).

The potential reason for those different trends is due to the North Atlantic influence over the western basin (Page 13 Line 34).

**Referee #2**: «The paragraph on Page 7 Line 6-10 only describes how the SST seasonal cycle amplitude of ROM is different to (i.e. smaller than) MPI-ESM but it doesn't support the statement of the authors that ROM is better than MPI-ESM due to the higher resolution. Moreover, "ROM overestimates the SST simulated be MPI-ESM" should be rephrased because "overestimates" is often used while comparing with an observation. Please rewrite this part. »

**Response**: Thank you. We have rewritten the paragraph in the revised manuscript (Page 8 Line 9-11).

**Referee #2**: «A comparison between SST time series of MPI-ESM with OISST (Figure 6) or an analysis about temporal correlation between MPI-ESM and OISST (Figure 7 & 15) make no sense as MPI-ESM doesn't know anything about SST of a certain 'real' year. Please remove these figures or at least the part of MPI-ESM and focus more on other results. »

**Response**: Thank you. We have removed the MPI-ESM results from the Figs. 6, 7, as well as Fig. 15 has been deleted in the revised manuscript.

**Referee #2**: «Section 2: details of basis configuration of REMO, MPI-OM such as horizontal and vertical resolution as well as running time step should be described. It's also necessary to give a list of variables exchanged between REMO and MPI-OM via OASIS. »

**Response**: We have added a new Table 1 where the characteristics of ROM atmosphere-ocean regional coupled model has been summarized. Furthermore, we have included the list of variables which exchange info between REMO and MPI-OM via OASIS in section 2.1 (see Page 3 Line 24-27 and Fig. 2.).

**Referee #2**: «Introduction: Page 2 Line 10-27 simply listed the previous study without any results mentioned. I suggest to summarize this part of introduction and mention more details about only the previous studies which gave information of climate change signal in SST and SSS that relates to the main topic of the current study. The authors should make it clearer what has been done in the past, what is still missing and why this study is important. »

**Response**: Thank you for the suggestion. We have reviewed the introduction and made reductions of previous works without results. Moreover, we have updated the introduction with recent information of climate change signal in SST and SSS in the revised manuscript (Page 2 Line 16-28).

**Minor Comments**

**Referee #2**: «Should it be "AORCMs" for coupled "atmosphere-ocean regional climate models" as used in many previous studies instead of RAOCMs? »

**Response**: Ok, we have used the acronym AORCMs instead of RAOCMs in the revised manuscript.

**Referee #2**: «Page 1 Line 24 is repeated at line 27. »

**Response**: The sentence (Page 1 Line 24) has been deleted in the revised manuscript (see section 1).

**Referee #2**: «Page 2 Line 34 & page 3 Line 4 have the same typo of "Asses" »

**Response**: Thank you. We have corrected the typos in the revised manuscript.

**Referee #2**: «Please follow the citation rule of the journal. At several places, "el al., (20xx)" was used where either "," or "(" is needed. »

**Response**: Thank you. We have made the correction in the revised manuscript.

**Referee #2**: «Page 2 Line 22: "Finally, Sevault et al. ...": does it mean it's the last fully coupled regional climate system model has been developed? »

**Response**: No, we just used "Finally" as a connector in the paragraph. Fully coupled models such as COSMO-NEMO_MFS (Cavicchia et al. 2015) or COSMO-CLM v4.21 (Akhtar et al. 2017) have been developed later than CNRM-RCSM4 (Sevault et al. 2014). For an updated revision of regional climate system models used in the Mediterranean region see Somot et al. (2018).

**Referee #2**: «Page 2 – Line 30 – Abbreviation MPI-OM is only introduced on page 3, Sect. 2. Please remove here. The abbreviation ROM has to be described as it is mentioned the first time in the main text (the abstract does not count). »

**Response**: We have made the correction in the revised manuscript.

**Referee #2**: «Sequence of 2.1 and 2.2 should be switched as REMO was mentioned before MPI-OM at the beginning of Section 2 Methods and also in sequence of abbreviation "AORCM". The author should also think about the sequence of "ocean-atmosphere" or "atmosphere-ocean" in the title to ensure the consistence for the whole manuscript. In addition, model abbreviations and references are provided in Sect. 2, and, hence can be removed from Sect. 2.1 and 2.2. »

**Response**: We have switched the sequence of 2.1 and 2.2 according with the abbreviation AROCM. The model abbreviations and reference have been removed from section 2.1 and 2.2. Furthermore, the sequence of ocean-atmosphere has been replaced for atmosphere-ocean in the title.

**Referee #2**: «Page 3 Line 9: For this work, the ROM climate model (Sein et al. 2015) has been used. »

**Response**: We have made the correction in the revised manuscript.

**Referee #2**: «Page 3 Line 12- ..., the soil model of REMO (Rechid ... »

**Response**: We have made the correction in the revised manuscript.

**Referee #2**: «Page 3 Line 13: Which version of OASIS was used? »

**Response**: OASIS version 3.0 was used.

**Referee #2**: «Page 3 Line 30: "REMO's prognostic variables are ...": for what are they important to be mentioned here? More important should be which variables are exchanged between REMO and MPI-OM via OASIS. »

**Response**: We have replaced the list of REMO's prognostic variables for the variables which exchange info between REMO and MPI-OM via OASIS (Page 3 Line 24-27, see Fig. 2).

**Referee #2**: «Page 4 Line 14: double "with" »

**Response:** We have made the correction in the revised manuscript.

**Referee #2**: «Page 4 Line 16: ... with a resolution of about 25 km ... »

**Response:** We have made the correction in the revised manuscript.

**Referee #2**: «Page 4 Line 18: why different coupling time steps (3 hrs & 24 hrs) are using? »

**Response:** For river runoff we do not need to reproduce diurnal cycle. Nevertheless, the REMO-MPI-OM coupling frequency is 60 min instead of 3h (it was a mistake in the old manuscript). It has been corrected in the revised manuscript (see Page 4 Line 23 and Table 1).

**Referee #2**: «Page 4 Line 21: "... scenario were analysed." »

**Response:** We have made the correction in the revised manuscript.

**Referee #2**: «Page 5 Line 18: ...of ROM's potential to improve the ... »

**Response:** We have made the correction in the revised manuscript.

**Referee #2**: «Page 6 Line 8: "can be found in DJF"? »

**Response:** We have made the correction in the revised manuscript.

**Referee #2**: «Page 6 Line 21: deviation of 3.5 mm/d is not a small amount. This corresponds to ~315 mm/season. Please comment more thoroughly! »

**Response:** We have rewritten this part more thoroughly in the revised manuscript (Page 7 Line 3-5).

**Referee #2**: «Page 7 Line 3: ... than expected ... »

**Response:** We have made the correction in the revised manuscript.

**Referee #2**: «Page 7 Line 5: ... have been done... »

**Response:** We have made the correction in the revised manuscript.

**Referee #2**: «Page 8 Line 10: "It is clearly seen how ..." »

**Response:** We have made the correction in the revised manuscript.

**Referee #2**: «Page 8 Line 11: "penetrate to the Western Mediterranean by the African continent": what do you mean? »

**Response:** We mean that the Atlantic inflow jet run close to the African continent when it goes to the Western Mediterranean. The sentence has been corrected in the revised manuscript (see Page 9 Line 8-10).

**Referee #2**: «Page 8 Line 14-15: sentence is incomplete. »

**Response:** We have made the correction in the revised manuscript (Page 9 Line 11-13).

**Referee #2**: «Figure 2: HD is missing. How are u & v surface currents passed from MPI-OM used in REMO? »

**Response:** We have included HD model in the scheme of ROM (see Fig. 2).

In REMO calculating turbulent fluxes we use relative surface winds, i.e. wind velocity minus ocean surface velocity (for details see Sein et al. 2015).

**Referee #2**: «Figure 6, 7 & 15: please remove the MPI-ESM_LR and MPI-ESM_MR as their SST temporal timeseries have non-sense. »

**Response:** We have removed the MPI-ESM results from the Figs. 6, 7, as well as Fig. 15 has been deleted in the revised manuscript.

**Referee #2**: «Figure 10: vector is too small. Why level 31m depth was chosen to be shown here? »

**Response:** We have chosen level 31m to remove the high-frequencies variability generated by the atmosphere (e. g. wind). Thus, allows us to represent averaged behavior of the surface Mediterranean circulation without altering as in previous works (L'Hévéder et al 2013; Sevault et al. 2014).

We have made bigger the vectors in Fig. 10.

**Referee #2**: «Figure 12: why do not show figures for DJF and JJA separately? A strong SST bias in summer of ROM (Fig.4) may affect the trend analysis if it's not system bias. »

**Response:** We have included figures for DJF and JJA as supplementary material. The trend analysis is not affected by the SST bias (Page 10 Line 12-14), especially in summer where ROM had shown a strong cold bias into whole Mediterranean Sea (see supplementary figures).

**List of relevant changes:**

- We have reviewed and updated the introduction with recent information of climate change signal in SST and SSS.

- We have included differences between ROM (coupled) and stand-alone REMO forced by ERA-Interim (see Fig. 4) with the aim to analyze the role of the interactive ocean on the REMO's atmosphere simulation.

- We have fully updated the discussion with the relevant literature from the recent Med-CORDEX activity.

**List of relevant changes:**

**The climate change signal in the Mediterranean Sea in a regionally coupled atmosphere-ocean model**

Ivan Parras-Berrocal[1], Ruben Vazquez[1], William Cabos[2], Dmitry Sein[3,4], Rafael Mañanes[1], Juan Perez-Sanz[2], Alfredo Izquierdo[1]

[1]Applied Physics Department, University of Cadiz, Cadiz, 11510, Spain
[2]Department of Physics and Mathematics, University of Alcala, Alcala de Henares, 28801, Spain
[3]Alfred Wegener Institute for Polar and Marine Research, Bremerhaven, 27570, Germany
[4]Shirshov Institute of Oceanology, Russian Academy of Science, Moscow, Russia

*Correspondence to*: Ivan M. Parras-Berrocal (ivan.parras@uca.es)

**Abstract.** We assess the climate change signal in the Mediterranean Sea with the regionally coupled model REMO-OASIS-MPIOM (ROM). The ROM oceanic component is global with regionally high horizontal resolution in the Mediterranean Sea. In our setup the Atlantic and Black Sea circulations are simulated explicitly. Simulations forced by ERA-Interim show a good representation of the present Mediterranean climate. Our analysis of the RCP8.5 scenario driven by MPI-ESM shows that the Mediterranean waters will be warmer and saltier across most of the basin by the end of the century. In the upper ocean layer temperature is projected to have a mean increase of 2.73°C, while the mean salinity increases by 0.17 psu, presenting a decreasing trend in the Western Mediterranean, opposite to the rest of the basin. The warming initially takes place at the surface and propagates gradually to the deeper layers.

**1 Introduction**

The Mediterranean Sea is expected to be among the world most prominent and vulnerable climate change "hot spots". In this context, climate change lies at the heart of sustainable development in the Mediterranean. As such, the region is an optimal test bed for new approaches to science-society partnership sustained by the provision of adequate climate information and applicable to a broad range of vulnerable sectors. The Mediterranean is a regional sea circumscribed by Africa, Europe and Asia and divided into two sub-basins (eastern and western) through a sill that does not exceed 400 m depth between Sicily and the African continent. The freshwater balance in the Mediterranean basin is negative, since the evaporation exceeds rainfall and river run-off (Sanchez-Gomez et al., 2011). This deficit is compensated by a net inflow of water through the Strait of Gibraltar. The region is located in a transitional area between tropical and mid-latitudes and presents a complex orography and coastlines where intense local air-sea and land-sea interactions take place. These intense local air-sea interactions together with the inflow of Atlantic water drive the Mediterranean thermohaline circulation (MTHC) (Fig. 1). For these reasons, atmosphere-ocean regional coupled models (AORCMs) are essential for the study of atmospheric and oceanic processes in the Mediterranean Sea.

To date, different AORCMs with typical horizontal resolution of 25-50 km in the atmosphere and 10-20 km in the ocean have been developed to study the climate of the Mediterranean Sea (Somot et al., 2008; L'Hévéder et al., 2013; Sevault et al., 2014; Cavicchia et al., 2015; Darmaraki et al., 2019). However, Akhtar et al. (2018) found the higher horizontal resolution (9 km) in the atmosphere improves the simulation of the wind and the turbulent heat fluxes, although they conclude that a higher resolution models do not perform better in all aspects than coarser configurations. Somot et al. (2008) developed the Sea Atmosphere Mediterranean Model (SAMM), which meant a new concept of AORCMs, composed by the coupling of the atmospheric global model (ARPEGE) (Déqué and Piedelievre, 1995) and the regional high-resolution (10 km) ocean model (OPAMED; Somot et al., 2006). Their results under the A2 (IPCC, 2000) climate change scenario showed an increment at the end of the 21$^{st}$ century of temperature and salinity both in shallow (3.1ºC and 0.48 psu) and in deeper layers (1.5ºC and 0.23 psu) of the Mediterranean Sea (Somot et al., 2006). In 2013 the European CIRCE project was launched (Gualdi et al., 2013), in order to ease the coordination among the scientific community responsible for regional climate modeling in the Mediterranean. The beginnings of CIRCE was can be traced to the work of Dubois et al. (2012) who compared different AORCMs and regional climate models (RCMs). In addition, these authors analyzed a projection (1950-2050) of the Mediterranean climate under the A1B scenario simulated by an ensemble of five coupled regional models. For the first time, atmosphere-ocean realistic net flows were obtained that predict a Mediterranean surface warming between +0.8ºC and +2ºC. Shaltout and Omstedt (2014) analyzed the Mediterranean SST for the 2005-2100 period projected by ensembles of CMIP5 (Taylor et al., 2012) global models under the RCP2.6, RCP4.5, RCP6.0 and RCP8.5 scenarios. The CMIP5 ensembles means indicate a warming, which ranges from +0.5ºC in the RCP2.6, through +1.15ºC in the RCP4.5, +1.42ºC in the RCP6.0, to +2.6ºC in RCP8.5 scenario. The authors conclude that the warming is mainly controlled by the amount of greenhouse gas emissions. More recently, Adloff et al. (2015) found that the mean Mediterranean SST and SSS will increase between +1.73 and +2.97ºC, and +0.48 and +0.89 psu at the end of the 21$^{st}$ century. Their results were based on an ensemble of six simulations performed with different configurations of the NEMOMED8 (Beuvier et al., 2010) ocean model under different scenarios. Darmaraki et al. (2019) employed an ensemble of 17 fully coupled atmosphere-ocean simulations to study the evolution of SST and marine heat waves in the Mediterranean Sea for 1976-2100 period. The ensemble means by the end of the century indicates a +3.1ºC increase of Mediterranean mean SST under the RCP8.5 scenario. By 2100, as a response of the sea surface warming, projections showed stronger and more intense Mediterranean marine heat waves. Most of these authors agree that the choice of emission scenario is the most important conditioning for the expected warming of the Mediterranean Sea.

These modeling efforts are coordinated through the Med-CORDEX initiative (Ruti et al., 2015; www.medcordex.eu), which is the regional climate modelling taskforce of the HyMeX program (www.hymex.org). In these models the oceanic component of the RAOCMs is also regional. The use an oceanic global model (MPI-OM) in REMO-OASIS-MPIOM (ROM) coupled system model, could help to avoid some problems associated with the open boundary conditions for the Mediterranean Sea, allowing to study processes that take place in the Mediterranean region but which have its origin at the North Atlantic Ocean. This work aims to contribute to the Med-CORDEX initiative with a first detailed evaluation of highresolution atmosphere-ocean simulations with the coupled ROM model, which has been used for some previous multimodel studies (see e.g. Darmaraki et al., 2019). Here we analyze the evolution of Mediterranean Sea under the RCP8.5 scenario with boundary conditions taken from CMIP5 simulation with the MPI-ESM global model. Especially, we focus on water masses properties such as SST and SSS and their evolution at the end of 21$^{st}$ century.

5    The objectives of this study can be summarized as follows:

[revised manuscript text omitted]

In this work, 30-year time series from three different experiments have been analyzed. The first simulation, thereafter
25    ROM_P0, was forced by ERA-Interim for 1980-2012 and used to assess the skills of ROM in reproducing the observed regional climate over the Mediterranean Sea. In order to offer an integrated vision of the impact introduced by the climate change in the Mediterranean Sea, we make a dynamical downscaled of present time simulation with MPI-ESM-LR which covers 1950-2005 period (for our analysis we take from 1976-2005, ROM_P1) and a climate change projection from 2006-2099 (for our analysis we take from 2070-2099, ROM_P2) under the Representative Concentration Pathways 8.5 (RCP8.5)
30    scenario.

[revised manuscript text omitted]
 have a larger dependency on the internal details of the atmospheric component, and the impact of the coupling is dependent on the large-scale circulation and land-sea contrasts. Therefore, we can expect the differences over

10 land to be overly small, except for the regions where the large-scale circulation or the land-sea contrasts are important.

The winter MSLP over the Atlantic is higher in the coupled run (Fig. 4a), causing an anomalous strong anticyclonic circulation that extends to land and the Mediterranean Sea, west of the Balearic Islands. The large-scale influence of the Atlantic anomalous circulation offsets the effect of the warmer SST here (see Fig. 5, where the SST biases are represented). However, elsewhere over the Mediterranean Sea, where the ROM SST is colder (warmer) than ERA-Interim, a higher

15 (lower) MSLP is simulated by ROM. In summer (Fig. 4b), the differences in MSLP seem to be determined mainly by the colder SST in ROM, which leads to higher MSLP in the model than in the reanalysis.

The changes in T2m induced by the coupling over the Mediterranean (Figs. 4c and 4d) seem to be mainly determinate by the SST, through the turbulent heat fluxes. In both seasons the T2m differences induced by the coupling correspond very well with the SST biases with respect to ERA-Interim. However, in winter T2m seems to be also influenced by the transport of

20 Atlantic air carried by the anomalous anticyclonic circulation simulated in the Atlantic. Over land the differences in winter T2m are mainly determined by the changes induced in large scale circulation by the interactive SST in the Atlantic, while in summer the land-sea contrasts seem to be more important.

The differences between the SST from ERA-Interim and the simulated by ROM is also reflected in the rainfall simulated by REMO and ROM (Figs. 4e and 4f). In winter the Mediterranean Sea regions where the ROM SST is warmer have a higher

25 precipitation, while colder ROM SST leads to a lower precipitation. The impact of the SST biases on the precipitation is clearer in summer: the cold SST bias leads in ROM to a precipitation which is weaker than in the REMO simulation all over the Mediterranean Sea, especially in the northern part, where the reduction of precipitation in ROM with respect to REMO is comparable in magnitude to the ROM precipitation bias (Fig. 3f).

**3.2. SST**

30 ### 3.2.1 Seasonal cycle

[revised manuscript text omitted]

30   JJA, a performance similar to other models (see e.g. Giorgi and Lionello, 2008). Positive MSLP biases over a large extend of the domain during DJF (Fig. 3a) could generate anticyclonic conditions which lead a greater stability and lower storm generation; while in JJA (Fig. 3b) the biases are generally much lower. With respect to the seasonal cycle of nearsurface atmospheric parameters such as near-surface (2m) temperature (T2m) and precipitations, the LMDz-NEMO-Med coupled model (L'Hévéder et al., 2013) gives a bias range of (-4; +4ºC/-2; +3 mm/d, respectively) which is comparable to the ROM estimates (Figs. 3c, 3d, 3e and 3f). Similarly, to most of Mediterranean regional models, ROM shows a higher than observed rainfall over areas with pronounced topography such as the Alps (Artale et al., 2010; L'Hévéder et al., 2013; Di

5 Luca et al., 2014). More recently, Fantini et al. (2016) also reported a similar bias (±3 mm/d) in an ensemble of regional coupled models forced by ERA-Interim. Panthou et al. (2018) observed that for heavy precipitation increasing resolution increases the wet biases. We totally agree with the final consideration of Fantini et al. (2016), which proposed that to improve the performance RCMs simulating precipitation it is necessary the availability of high quality and high-resolution observation for the assessment of the models.

10 The comparison of the ROM with standalone REMO shows that the changes in SST generated by the coupling in the Atlantic Ocean influence the simulated Mediterranean climate, causing a spurious anticyclonic circulation in winter which impacts the surface temperature in the Western Mediterranean. In summer the modeled SST is significantly colder than observations, leading to colder T2m and less precipitation over the basin, as the colder SST reduces the evaporation.
Regarding the oceanic parameters, ROM shows biases within ±3ºC, correlation coefficients above 0.7 and RMSE below

15 0.25ºC when compared to ERA-Interim, EN4 and OISST data sets. ROM presents cold biases along the Eastern Mediterranean that become stronger and extend to the whole basin in summer months. The summer biases are common to most of the Mediterranean regional coupled simulations (see for instance, Dubois et al., 2012; Li et al., 2012, Sevault et al., 2014). Akhtar et al. (2018) studied the impact of resolution and coupling in modelling the climate of the Mediterranean Sea and concluded that coupling generates a negative bias in SST. Most recently, Darmaraki et al. (2019) assessed an ensemble

20 of 17 simulations from six models, in which our ROM coupled system was included. Their results showed an averaged cold bias ranging from (-0.29 to -1.01ºC) when regional models are compared to satellite data. When forced by ERA-Interim, ROM shows averaged Mediterranean SSTs that are colder than reference climatologies during 1980-2012 period (Fig. 7), a common trait with other RCSMs (Sevault et al., 2014; Ruti et al., 2015). Macias et al. (2018) showed that a simple spatially-uniform bias correction improves the simulated surface oceanic conditions of the Mediterranean basin when forcing an

25 oceanic model with atmospheric data from RCM realizations. These results show the summer biases could be related either to a deficit of solar radiation by the atmospheric model or to shortcomings in the simulation of some processes in the ocean model, as vertical mixing or turbidity. As theses biases appear in the coupled runs, we could speculate that some coupling feedbacks are present. This topic deserves a separated study and will be tackled in a future paper. The SSS simulated by ROM shows seasonal biases in the -1; +1 psu range, with a similar magnitude and spatial distribution than those in RCSM4

30 (Sevault et al., 2014). The biases are higher in some problematic locations such as the northern Adriatic Sea and Dardanelles Strait (Fig. 9), a feature that also has been shown in previous studies (L'Hévéder et al., 2013; Di Luca et al., 2014; Sevault et al., 2014). The Mediterranean water fluxes simulated by ROM (Table 4) have been compared to available observations (Sanchez-Gomez et al., 2011; Soto-Navarro et al., 2014) and model (Sevault et al., 2014) estimates, providing a physically consistent assessment in the straits. ROM water balance terms over the Mediterranean Sea are similar to those obtained by

different authors (Table 4). The main difference is the exchange flows through the Strait of Gibraltar, where ROM presents estimates much lower than those presented by Soto-Navarro et al. (2014), although the net flow is in agreement with most of estimates.

The ROM SSH and surface (31m) circulation are able to reproduce the different quasi-permanent elevation/depression (anticyclonic/cyclonic) structures occurring in the Mediterranean Sea (Fig. 11). The cyclonic gyres (SSH depressions) correspond properly to the water mass formation sites. The 31m depth level has been chosen to remove the high-frequency variability of the uppermost ocean, while retaining a characteristic upper ocean circulation pattern. Also, the choice of this level depth makes our result more comparable with other works such as L'Hévéder et al. (2013) and Sevault et al. (2014). For 1980-2012 period the comparison between ROM and AVISO (SSALTO/DUACS, 2013) altimetry data (Fig. 12a) produced a quite satisfactory correlation of 0.61, close to the obtained by the RCSM4 (0.68). Finally, the ROM amplitude of the mean seasonal cycle measured was 12 cm while for AVISO was 14.5 cm and for RCSM4 16.9 cm (Fig. 12b). Thus, the model is able to satisfactorily reproduce the seasonal cycle and interannual variability of different oceanic variables.

The model also demonstrated good skills in reproducing the area-averaged interannual standard deviations of SST for the Mediterranean Sea (Fig. 16d). According to Fig. 16, ROM coupled system presents yearly SST standard deviations close to the reference OISST dataset. In fact, ROM does not only improve the yearly spatial standard deviations with respect to the MPI-ESMs (Figs. 16e and 16f) but also regarding to ERA-Interim and EN4 (Fig. 16b and 16c). The MPI-ESM-LR and -MR are not able to reproduce those local patterns due to the absence of resolution, thus indicating that the dynamical downscaling from MPI-ESM improves the simulation of GCMs.

In our simulations, the Mediterranean Sea will be warmer and saltier at the end of 21$^{st}$ century. Under the RCP8.5 scenario ROM provides integrated estimates of climate change similar to other models (Table 5). The mean ΔSST projected by ROM under RCP8.5 scenario is 2.73ºC, close to MPI-ESM simulations, which show an SST increase of 2.80ºC (-LR) and 2.87ºC (-MR). It is also close to the mean increase (+2.6ºC) projected by the CMIP5 ensemble of Shaltout and Omstedt (2014) (Table 5). The SST warming estimates under the RCP8.5 scenario agree with those obtained by Adloff et al. (2015) and with the multi-model Mediterranean of Darmakari et al. (2019), which show a mean increase of +3.1ºC, with model increases ranging from +2.7 to +3.8ºC. Despite differences horizontal resolutions, ROM expected warming (Fig. 13b) shows a similar spatial distribution to MPI-ESM-LR than MPI-ESM-MR (Figs. 14a and 14b). This is due to the ROM_P1 simulation used to computed the ΔSST (ROM_P1-ROM_P2) was forced by MPI-ESM-LR. Figs. 15a and 15b shown how the warming that initially takes place at the surface layer is transported gradually to deeper layers. The warming expected by the models and its intensity will be strongly linked with the choice of emissions scenario.

As shown in Table 5, the mean ΔSSS projected by ROM under RCP8.5 is lower than those estimated by other authors (Somot et al., 2006; 2008, Adloff et al., 2015). This seems to be related to the fact that the SSS filed in the ROM RCP8.5 projection shows a dipolar structure in the Mediterranean (Fig. 13d). So far, previous works (Somot et al., 2006, Adloff et al., 2015) denoted positive values of ΔSSS over the whole Mediterranean Sea by the end of the century, which differ to our results. The ROM 
[revised manuscript text omitted]

---

## Referee Report (RR1)

Comments for the revised manuscript **"The climate change signal in the Mediterranean Sea in a regionally coupled ocean-atmosphere model"**

- The first comment of the Referee#1 touched an important point about a projection run with the ocean driven offline by stand-alone REMO in the coupling area. However, the authors provided the new figure 4 of differences between ROM_P0 and the stand-alone REMO forced by ERA-Interim. Here we can see a different/opposite behaviour of ROM_P0 and the stand-alone REMO. For example, for T_2M in JJA over the Mediterranean Sea, ROM_P0 is about 1-2°C colder than ERA-Interim (Fig.3d) but about 1.5 – 2°C colder than REMO (Fig.4d). Thus, the stand-alone REMO can be about 0.5°C warmer than ERA-Interim. I suggest the authors redo the figure 3 for the stand-alone REMO. In case the opposite results are found, I think it is interesting to see how the projection for temperature, salinity and sea level height will be when the stand-alone REMO is used over the coupling domain, if these experiments are available. If the authors have not done these experiments yet, I suggest to write a sentence about the planned experiments as an outlook for the future work.

- Abstract: "We assess the climate change signal in the Mediterranean Sea with the regionally coupled model": Shall it be "provide" or another more suitable word instead of "assess"? The authors can compare the climate change signal obtained from your model results with other previous studies but cannot assess whether the signal is right or wrong because the truth is unavailable.

- Can the authors argue in the manuscript why you used the MPI-EMS_LR to force ROM instead of MPI-EMS_MR?

- The reference Somot et al. (2018) is not cited in the manuscript.

- Table 1: time step of REMO is missing.

- Figure 11: please re-plot the figure with a different scale of current velocity (e.g. 0.1 m/s) to make the vectors more visible. Zoom in the figure doesn't help to make the vectors more visible but decrease the quality of the figure.

- Page 2 Line 31: still "RAOCMs"

- Page 3 Line 15: It should be better with "For this work, the ROM climate model (Sein et al. 2015) has been used.". The current sentence "For this work, the ROM climate model has been used (Sein et al. 2015)." sounds like this current work was already published in Sein et al. (2015).

- Page 3 Line 24: "info" is an informal word. Please use "information".

- Page 5 Line 29: "For a better … in the Mediterranean Sea, comparisons …": The added "," would make the sentence easier to understand.

- Page 6 Line 28: "The largest discrepancies for DJF are located" or "… DJF can be seen …"

- Page 7 Line 1: is summer the very dry season in Mediterranean Sea region? or do you mean a specific very dry summer?

- Page 7 Line 7-9: Please rewrite the sentence: "Over land the simulated fields have a larger dependency on the internal details of the atmospheric component, and the impact of the coupling is dependent on the large-scale circulation and land-sea contrasts."

- Page 7 Line 12-13: "The large-scale … offsets the effect …": I do not get the meaning of this sentence.

- Page 7 Line 17: "determinate" → "determined"

- Page 7 Line 25-28: sentence is too long and not clear what the authors mean.

- Page 8 Line 10: "near to -1ºC": should use "approximately" or "about"

- Page 9 Line 10: "inflow jet runs along the African continental coastline"

---

## Referee Report (RR2)

**Comments on the paper „The climate change signal in the Mediterranean Sea in a regionally coupled atmosphere-ocean model"**

**General comments**

The paper describes simulations with a coupled ocean-atmosphere model with the interesting feature that the ocean is in a global set-up coupled to a regional atmospheric model. The analyses are interesting and the paper is generally very interesting for the scientific community.

However, the paper needs major revisions before it can be published. The English has to be checked by someone professional, at the moment the paper is full of strange formulations and errors. It is unclear to me, why the chapters 3 lack almost completely citations, except 3.4, and all the relevant point and citations are to be found in chapter 4. It would be easier to read when the results are compared to other scientific publications and in the discussion, the processes are discussed. My annotations are full of "whys" and almost no answers can be found.

There is no explanation about the initialization of the ocean modell. Was there a spin-up calculated and if yes how? Usually MPI-OM needs a coupled of hundred years to reach quasi-equilibrium. This is the big disadvantage of a model setup like yours so it should be discussed.

Please use ROM_P0 or P1 everywhere in the text where you discuss your runs and not the model in general. It is less confusing then comparing the discussions with the figures, where these names are used. Please label the figures itself with DJF or SST or whatever, so it is faster to grasp what is shown.

In detail:

Page 1, line 13: the Black Sea is mentioned already in the abstract but no results regarding climate change variability w.r.t. the Black Sea is discussed. And no results at all about the Black Sea are shown, not to speak of the circulation as mentioned here.

Line 20: Please add citation for hot spot. The 2$^{nd}$ sentence of the Introduction is mere speculation without any base.

Line 26f: the deficit is compensated by the Strait of Gibraltar. Does this mean the inflow of fresh water from the Black Sea does not play a role in the budget?

Line 28: local intense air-sea interaction does not make sense. Only if the whole MedSea is local, which is kind of strange in this paper.

Page2, line 12: pls skip the "was".

Line 27f: This is also a strange perspective. It is not the choice of the scenario which conditions the signal, it is driving factors which are prescribed in the scenario. And the scenarios are well thought off pathways of the future evolution of the climate change signal. This sentence here can be understood by climate critics that you only have to chose the right scenario to get the answer you want. This should not be written like this.

Line 32f: what are the problems with open boundary conditions you mention here? This is one of the key features of your model setup so please describe it with more insight.

Line 34: it is the first detailed evaluation but the run has also been used in other studies? A bit of a contradiction here.

Page 3: line 4: evolution at the end? Maybe towards the end would be better. It is a process.

Line 6: skill not skills

Line 8: better: driving model (mpi-esm) and skip the last part of the sentence.

Line 24: info*rmation*

Page 4, line 27: "we make a dynamical downscaled of present time simulation". Please rewrite this sentence. Why did you force the ROM with MPI-ESM-LR? Later it is getting clear, that LR and MR differ very much although it is nowhere discuss why, so why this forcing?

Page 5, line 23: "we made comparisons". No good english.

Line 25: based on the NEMO code. Which model do you talk about here. Generally, I would skip a lot of information on the data but the resolution in space and time and the citations. But this is your choice.

Line 30: against *other* ESMs are required.

Line 30: Why and how is the setup of MPI-OM different compared to MPI-ESM? And how are these differences relevant when looking at the results of ROM compared to the MPI-ESM-LR/MR?

Page 6, line 1: is HAMOCC used? What role does it play in the MedSea climate change simulations? Why is OASIS3 used and not OASIS3-MCTx?

Line 17f: what is the explanation for the overestimation of the azores high? The location is quite close to the boundary of REMO so where does it come from? The boundary formulation? If the ocean is the source, then there are deficits in the ocean circulation in the Atlantic which might play a role in the MedSea as well.

Line 23f: Could it be that REMO has a problem simulating the circulation over mountainous areas? There used to be a formulation in REMO smoothing out this effect, maybe this was not used? Generally, the great benefit of regional climate modelling is the higher resolution accompanied by a better representation of mountains. There is a sentence (Page 7, line 9f) in the paper talking about the benefits but leaving out this mountain/ororgraphy effect. So please discuss this point somewhere.

Line 26: could play a role. Good question. Do they? Did you look at this point in more detail? Or is it due to interpolation artefacts?

Line 30: Same is true for the last sentence on this page, did you check where the anomalies come from?

Page 7, line 11ff: Why is MSLP higher in the coupled run?

Page 8, line 2: more the northern part of the eastern MedSea.

Line 6: higher resolution where ocean or atmos?

Line 8: what simulation is discussed here? P0?

Line 17f: what data set configuration?

Line 20: this is the first time, aerosols are mentioned. It would be nicer to have this information in the general description of the model. And then, why are aerosols neglected?

Line 21: offset of SST: what about the spin-up of the ocean model?

Line 25f: last part of the sentence is not understandable. And what is climate uncertainty? Please define.

Line 27: Chapter 3.3: The influence of HD is nowhere discussed. Why is there so much freshwater inflow from HD? This is missing in this chapter.

Page 9, line 3: why is ROM always saltier? HD? Spin-up? Surface fluxes? Please explain.

Line 29: RCSM4 model turns up out of the blue. What model is this, citation?

Page 10, first paragraph: global warming everywhere? Please skip such general sentences.

Line 13: Figure 13 is not easy to understand, so please label this figures themselves with variable and time slot.

Line 6f: which run is discussed, which period. And an av. Zonal SST from north to south is really hard to distinguish from that figure.

Line 21: already in the present day simulation is too much fresh water in the Bosporus and close to the Nile, this will be transported to the future scenarios.

Line 26f: where does the diff between MR and LR come from?

Line 19: where did you discuss the circulation in the Adriatic Sea and compared it to observations?

Page 12, first paragraph: the discussion would be a bit more interesting if the resolution of the different models mentioned compared to ROM would be written somewhere.

Line 27f: so why do you speculate and not do some analysis and answer this question?

Page 13, line 4: please mention the 31m in the model description. Is this the depth of the first layer or only the choice to compare it to other studies? Unclear.

Line 11: RCSM4 is not to be found in Fig 12b.

Line 17: what is the absence of resolution? Sound more like a philosophical question.

Line 24: what is the multi-model Mediterranean? And what are model increases? Lot of errors on this page. ☹ Like differences instead of different and filed instead of field and so on. Not very nice to read. Lots more.

Page 14, line 9: where was the transport through the Dardanelles Strait discussed?

Generally: in the conclusion: I couled like to see a good overview about the benefits of this coupled system ROM compared to others without the global ocean, a short overview over the evaluation, i.e. how well is the model behaving in the ERA-Interim simulation to have a good feeling about the future projections. And then the climate change signal which was simulated. Maybe some problems/shortcomings of the model and some ideas how to improve well known deficits.

---

## Referee Report (RR3)

**Comments on the paper „The climate change signal in the Mediterranean Sea in a regionally coupled atmosphere-ocean model"**

**General comments**

The paper describes simulations with a coupled ocean-atmosphere model with the interesting feature that the ocean is in a global set-up coupled to a regional atmospheric model. The analyses are interesting and the paper is generally very interesting for the scientific community.

Still there are many open points to discuss. After re-reading the article Sein 2015, I would like to know how it is possible to give climate change signals of salinity with a precision of two place after the comma? What about salinity restoring? How is this done in this setup and especially in the climate projections? And what is the influence of the North Atlantic on the circulation in the MedSea? And is it important to simulate the Strait of Dardenelles explicitly or is this irrelevant for the circulation in the MedSea? And what are the transports in the Strait of Gibraltar in the spinup and then later in the climate predictions? Is it changing a lot due to the global ocean?

And in the discussion, you mentioned other articles which use bias correction. Then why didn't you do this? This would probably give better relative numbers of SST and SSS changes due to climate change.

Please check the English by a native speaker, there are still some errors in the paper. It would be easier to read.

**Comments in details**

Abstract: Please skip the second place after the comma altogether, no model is that precise. It gives a false sense of trust.

Line 19: skip the first "the" and it is water instead of waters.

Introduction:

Page 3, Line 21: ROM does not add anything to the driving model. Maybe add "compared to" after the adds or ask a native speaker what would be a better formulation.

Page 4, Line 11: (1984). Here, the spatial….

Line 12: parametrization in the atmospheric component.

Page 5, Line 2: first layer nominal thickness of 8m.

Line 4: until an quasi-equilibrium is reached. Question: which parameter did you investigate to judge whether an quasi-equilibrium is reached?

Line 5: there is a forlorn "e" in this sentence. And please add, which period you repeat with the coupled system (1979-?).

Line 33: behave similarly

Line 34: mean state and its variability.

Page 6, Line 4: instead of some write: of several atmos. and oceanic variables.

Line 8: are derived. Question: What? You derive your own observational data set from whatever is appropriate? Please rephrase in a scientific way.

Line 22: OISST is according to you the best/observed dataset available. Could you show and proof this? And if not could you add a citation? Otherwise, this is hearsay.

Page 7: Section 3.1. Here and ERA-Interim highres simulation is compared to an ERA-Interim lowres simulation. So all the difference should come from the resolution differences, from different physics formulations and parametrizations, or from boundary effects. In Sein (2015) one conclusion is that the Gulfstream separation is improved in that setup. Why is this not the same in your simulations? Where do the differences in MSLP come from? Boundary effects in the atmosphere? This would be a disadvantage of your setup.

Line 20: "may play a role". Please write as a scientist. Either you checked and/or did some experiments or you don't know then you check in the future.

Line 23: I don't see a distinct signal southwest of the IberianPeninsula.

Page 8, Line 5-6: which formulation might this be?

Line 12: colder SST in ROM_P0. Question: why is the SST colder and what is the longterm change in the coupled system during the different spin-ups?

Line 22: So it should rain a lot over the Black Sea, which is not the case. Maybe the explanation is a bit too simple?

Line 24-25: which is not true for the Adriatic Sea.

General Question to section 3.1: there seem to be a strong signal at the eastern boundary of REMO (see Fig 3e,3f) in the same order of magnitude as the signal over the ocean, even stronger in summer. How do you discuss this effect? Is it a reflection at the boundary.

Page 9, Line 14: where is the 0.1 to be seen?

Line 18: how can the internal variability be properly reduced? How can this be done?

Line 23-24: The last sentence seems to be a bit farfetched. I would interpret it is outside the range of the uncertainty of gridded datasets.

Page 10, Line 1: I understand the runoff is smaller than the observed estimates. How does this match with the last sentence of this paragraph (line 4) "also larger than estimates"?

Line 2: I guess it should be SSS instead of SST?

Line 6-7: skip "the" before atmosphere and ocean (native speaker check would be good).

Line 11: Why is the time-averaged SSH compared to the velocity at 31m? The explanation is in the discussion but it should be here when first mentioned.

Line 19: which mesoscale structures are not represented? Should I cross check in the other paper? Somehow weird.

Page 11, Line 16: what is the contribution of the Dardanelles/Black Sea in the mean budget, in the variability and in the climate change signal? How well is the Black Sea represented and how important is it to include this side ocean in the coupled system? And please skip the second number after the comma. 38.02. Come on. Round it.

Line 25: What about the Dardanelles (again) seems also to ne pretty warm and salty.

Line 20: General question: why is it so much saltier in the eastern MedSea? Is the evaporation strongly enhanced in the east compared to the west? Does the intermediate water formation change so the vertical column is differently composed? Compared to Fig 15 I would conclude, the water at the surface gets saltier which is then transported in deeper layers. What is the variability of the intermediate and deep water formation regions (Adriatic and Levantine Sea)? And how does the surface circulation change?

Page 12, Line 2: most pronounced instead of more evident.

Line 7: significant comes in science with a significance test. Did you do one? If not change the wording.

Line 11: skip the "throughout the current century". And the "the" before temperature and salinity.

Line 15: In the first sentence, you cite Sein 2015 and the next sentence you talk about our model, which gives the impression, these are different models.

Line 18: Instead of "realisitic", write more realisitic. And Straits with capital S.

Line 20: no, there is no lack of resolution. What should it lack? Really wrong wording, skip it everywhere you ever wrote it. "Due to the coarse resolution" would be better.

Line 22: how can you lose spatial resolution? You could lose benefits or money but not resolution. Without the advantage of higher resolution maybe.

Line 23: Sein 2015?

Line 30: very old citation aren't there any more recent ones?

Page 13, line 1: to too cold modeld SST. Is this everywhere true? Seems to be a very general statement.

Line 7: when increasing the resolution you get more precipitation? The why in ROM_P0 saltier than MPIESM (Fig 10)?

Line 8: This argument is nonsense but also somehow true. To improve the precipitation in climate model, the parametrizations have to be improved or the convection has to be resolved in very high resolution. And to compare this changes we need urgently observation with a high temporal and spatial frequency. Only the data on its own will not improve any biases. We need the data to improve the models and then to evaluate them.

Line 9: in this strange sentence there is and "are" missing.

Line 10: Where is this shown, the reaction of ROM outside the MedSea region? This would really be interesting to be discussed.

Line 16: skip SSS, it is not discussed in this paragraph at all.

Page 14: Line 1: change would to could.

Line 10: why is the transport through the Straits of Gibraltar lower than other estimates although a global ocean model is used where the circulation is more realistic than in other AORCMs in this region?

Line 12: quasi/permanent. Is this also true for the future? What is changing?

Line 23: Where is the added value of ROM_P0 over other AORCMs?

Line 34: Again, something is lacking. And please change "improves" to "refines". And improvement would correct the deficits of MPI-ESM.

Page 15, Line 16: as already mentioned above, an LIW discussion would be nice.

Line 21: which benefits? Please skip the word "some" in the whole article. Completely unscientific.

Line 24-25: This setup would improve what? It does improve the…. And how is the ocean component adjusted? Very very strange sentence. Some for the next sentence: global ocean model improves something… without global ocean. Better: the use of a global ocean model could improve AORCMS who prescribed the global ocean as boundary condition.

Line 28: outside the coupled domain: not shown here.

Page 16, Line 4-5: Where is it shown that the exchange through the straits is improved? I understood at least for Gibraltar the transports are too low. And what internal behavior of which ESMs did you discuss? Where?

---

## Referee Report (RR4)

**Comments on the paper „The climate change signal in the Mediterranean Sea in a regionally coupled atmosphere-ocean model"**

**General comments**

The paper describes simulations with a coupled ocean-atmosphere model with the interesting feature that the ocean is in a global set-up coupled to a regional atmospheric model. The analyses are interesting and the paper is generally very interesting for the scientific community.

The authors have been very patient and thorough in answering all my comments, it was a pleasure to read the authors response. I have only the additional comment that some points were discussed very nicely in the response where I wished to have an additional sentence in the document itself, i.e. the connection between SST and precipitation over the Black Sea. Maybe other readers have the same questions as I had and would be glad for an additional sentence.

I have still a question mark when it comes to the bias correction. Same here, it would be nice to have a bit of your explanation in the paper:

```
„Macías et al. (2018) apply the bias correction to the RCM atmospheric fields used as
surface
forcing in an uncoupled simulation with their ocean model. We cannot apply their approach
to our
coupled system. "
```

Why?

```
„However, we have identified the process at large (better: that are?) responsible for
(the) ROM SST bias. We consider that this is also an important finding for improving ROM's
performance. "
```

In your conclusion you wrote that the found biases are in the range of other biases reported and
```
„However, there is place for further improvement in reducing certain biases (SST and MSLP)
by isolating the causes through targeted sensitivity experiments. "
```

So when I understand your two sentences above right, you know already the sources of the bias and you can eliminate them to improve ROMs performance. Thus it would be great to have a statement like this in your conclusions.

---

## Author Response (AR3)

We would like to thank the reviewer for the thorough review and the suggestions that have helped to significantly improve our manuscript "The climate change signal in the Mediterranean Sea in a regionally coupled atmosphere-ocean model". The point-by-point response to the review comments are presented in bold, whereas our answers are in standard text.

**Reviewer's comments:**

**Referee #2: The first comment of the Referee#1 touched an important point about a projection run with the ocean driven offline by stand-alone REMO in the coupling area. However, the authors provided the new figure 4 of differences between ROM_P0 and the stand-alone REMO forced by ERA-Interim. Here we can see a different/opposite behavior of ROM_P0 and the stand- alone REMO. For example, for T_2M in JJA over the Mediterranean Sea, ROM_P0 is about 1-2°C colder than ERA-Interim (Fig.3d) but about 1.5-2°C colder than REMO (Fig.4d). Thus, the stand-alone REMO can be about 0.5°C warmer than ERA-Interim. I suggest the authors redo the figure 3 for the stand-alone REMO. In case the opposite results are found, I think it is interesting to see how the projection for temperature, salinity and sea level height will be when the stand-alone REMO is used over the coupling domain, if these experiments are available. If the authors have not done these experiments yet, I suggest to write a sentence about the planned experiments as an outlook for the future work.**

Response: Thank you for the suggestion. We agree with both reviewers that a projection run with the ocean driven offline by stand-alone REMO in the coupling area would add valuable information on ROM behavior and we are planning to make such a run in the near future. We recognize the benefit of such an experiment and have added to the manuscript the following statement (see Page 13 Line 13f):

"In order to explicitly assess the role of the regional coupling on the simulated temperature, salinity and sea level, the results presented here will be compared with those from MPI-OM driven offline by stand alone in the coupling area, which is in progress."

**Referee #2: Abstract: "We assess the climate change signal in the Mediterranean Sea with the regionally coupled model": Shall it be "provide" or another more suitable word instead of "assess"? The authors can compare the climate change signal obtained from your model results with other previous studies but cannot assess whether the signal is right or wrong because the truth is unavailable.**

Response: Thank you for the suggestion. We have used "analyze" instead of "assess".

**Referee #2: Can the authors argue in the manuscript why you used the MPI-EMS_LR to force ROM instead of MPI-EMS_MR?**

Response: At the end of section 2.3 we now explain why we use MPI-ESM-LR as driving model instead of -MR:

"We have used MPI-ESM-LR to force ROM in experiments ROM_P1 and ROM_P2 because -LR was used in a wider set of CMIP5 experiments and with more realizations than -MR (Giorgetta et al., 2013). Both present the same horizontal resolution in the atmosphere, and although MPI-ESM-MR has a higher vertical resolution, mainly in the upper troposphere and lower stratosphere, the main differences in the simulations can be found in the middle atmosphere (Stevens et al, 2013). According to a recent benchmarking exercise of CMIP5 models (Lauer et al., 2017) their overall performance is quite similar. Jungclaus et al. (2013) provided a detailed description and evaluation of the ocean performance of MPI-ESM-LR and -MR, and concluded that both behaved similarly in many aspects, although -LR simulated the Labrador Sea and the North Atlantic more accurately, at least in the mean state and mean variability features."

**Referee #2: The reference Somot et al. (2018) is not cited in the manuscript.**

Response: Thank you for the remark. We have now included the reference (see Page 2 Line 34)

**Referee #2: Table 1: time step of REMO is missing.**

Response: Thank you for the remark. REMO integration time step is now indicated in the text and in Table 1.

**Referee #2**: Figure 11: please re-plot the figure with a different scale of current velocity (e.g. 0.1 m/s) to make the vectors more visible. Zoom in the figure doesn't help to make the vectors more visible but decrease the quality of the figure.

**Response**: We have followed the suggestion of the reviewer regarding the figure, choosing 0.1 m/s as scaling factor in order to improve the figure (the wide range of velocities makes it difficult to render a nice vector field).

**Referee #2**:  Page 2 Line 31: still "RAOCMs"

**Response**: We have made the correction in the revised manuscript.

**Referee #2**: Page 3 Line 15: It should be better with "For this work, the ROM climate model (Sein et al. 2015) has been used.". The current sentence "For this work, the ROM climate model has been used (Sein et al. 2015)." sounds like this current work was already published in Sein et al. (2015).

**Response**: We have made the correction in the revised manuscript.

**Referee #2**: Page 3 Line 24: "info" is an informal word. Please use "information".

**Response**: We have made the correction in the revised manuscript.

**Referee #2**: Page 5 Line 29: "For a better ... in the Mediterranean Sea, comparisons ...": The added "," would make the sentence easier to understand.

**Response**: We have made the correction in the revised manuscript.

**Referee #2**: Page 6 Line 28: "The largest discrepancies for DJF are located" or "... DJF can be seen ..."

**Response**: We have made the correction in the revised manuscript.

**Referee #2**: Page 7 Line 1: is summer the very dry season in Mediterranean Sea region? or do you mean a specific very dry summer?

**Response**: We mean that Mediterranean summers are very dry. We have tried to avoid misunderstandings in the manuscript (see Page 7 Line 27)

**Referee #2**: Page 7 Line 7-9: Please rewrite the sentence: "Over land the simulated fields have a larger dependency on the internal details of the atmospheric component, and the impact of the coupling is dependent on the large-scale circulation and land-sea contrasts."

**Response**: We have rewritten the sentence in the revised manuscript (see Page 8 Lines 1f).

**Referee #2**: Page 7 Line 12-13: "The large-scale ... offsets the effect ...": I do not get the meaning of this sentence.

**Response**: We have rewritten the sentence in the revised manuscript (see Page 8 Line 8f).

**Referee #2**: Page 7 Line 17: "determinate" ◊ "determined".

**Response**: We have made the correction in the revised manuscript.

**Referee #2**: Page 7 Line 25-28: sentence is too long and not clear what the authors mean.

**Response**: We have rewritten the sentence in the revised manuscript (see Page 8 Line 22f).

**Referee #2**: Page 8 Line 10: "near to -1$^{0}$C": should use "approximately" or "about"

**Response**: We have made the correction in the revised manuscript.

**Referee #2**: Page 9 Line 10: "inflow jet runs along the African continental coastline"

**Response**: We have made the correction in the revised manuscript.

**Referee #3**: **The English has to be checked by someone professional, at the moment the paper is full of strange formulations and errors. It is unclear to me, why the chapters 3 lack almost completely citations, except 3.4, and all the relevant point and citations are to be found in chapter 4. It would be easier to read when the results are compared to other scientific publications and in the discussion, the processes are discussed. My annotations are full of "whys" and almost no answers can be found.**

**Response**: Thank you for your remarks.

The English has been checked by a professional in the revised manuscript.

Regarding Sections 3 and 4: we consider it better to show ROM model results and its evaluation compared to the reference datasets and MPI-ESMs in section 3, while placing the results in a wider context (comparison to other scientific publications) and the discussion of processes in section 4, as they often complement or support each other. We have decided to maintain the same structure, but Sections 3 and 4 have been significantly rewritten, in order to improve the readability.

We hope your questions will be answered in this document and in the revised manuscript.

**Referee #3**: **There is no explanation about the initialization of the ocean model. Was there a spin-up calculated and if yes how? Usually MPI-OM needs a coupled of hundred years to reach quasi-equilibrium. This is the big disadvantage of a model setup like yours so it should be discussed.**

**Response**: The spin-up of MPI-OM was done according to the procedure described in Sein et al. (2015): In the stand-alone mode, MPI-OM is started with climatological temperature and salinity data (Levitus et al., 1998). Subsequently, it is integrated four times through the 1948-2000 period forced by ERA-40 until quasi-equilibrium is reached. For the coupled runs, the model is started from the final state reached in the last stand-alone run e integrated again three times forced by ERA-Interim (see Page 5 Line 2f).
Once the quasi-equilibrium state has been reached, it can be used as initial condition for different simulations around different areas, thereby alleviating the described disadvantage.

**Referee #3**: **Please use ROM_P0 or P1 everywhere in the text where you discuss your runs and not the model in general. It is less confusing then comparing the discussions with the figures, where these names are used. Please label the figures itself with DJF or SST or whatever, so it is faster to grasp what is shown.**

**Response**: We have used the names of each ROM simulations (P0, P1, P2) and labeled the figures throughout the revised manuscript.

**Referee #3**: **Page 1, line 13: The Black Sea is mentioned already in the abstract but no results regarding climate change variability w.r.t. the Black Sea is discussed. And no results at all about the Black Sea are shown, not to speak of the circulation as mentioned here.**

**Response**: The Black Sea and the Atlantic Ocean have been mentioned to describe the model domain and to stress the global coverage of the ocean model and that the water exchanges of the Mediterranean with the adjacent basins are not prescribed, but explicitly simulated by ROM. In this paper we do not aim to study the Atlantic Ocean or the Black Sea. In order to avoid any misunderstanding we have reformulated that sentence in the abstract to

"so that the water exchanges with the adjacent North Atlantic and Black Sea are explicitly simulated"

**Referee #3**: **Line 20: Please add citation for hot spot. The 2$^{nd}$ sentence of the Introduction is mere speculation without any base.**

**Response**: Thank you for the remark. We have added the citation for hot spot (Giorgi, 2006; Cramer, 2019). The 2$^{nd}$ sentence of the Introduction has been removed.

**Referee #3**: **Line 26f: the deficit is compensated by the Strait of Gibraltar. Does this mean the inflow of fresh water from the Black Sea does not play a role in the budget?**

**Response**: Thank you for the remark. Of course the water exchange through the Dardanelles plays its role in the Mediterranean Sea water budget. We have made the correction in the revised manuscript.

**Referee #3**: **Line 28: local intense air-sea interaction does not make sense. Only if the whole MedSea is local, which is kind of strange in this paper.**

**Response**: Thank you for the remark. We have made the correction in the revised manuscript.

**Referee #3**: **Page2, line 12: pls skip the "was".**

**Response**: We have made the correction in the revised manuscript.

**Referee #3**: **Line 27f: This is also a strange perspective. It is not the choice of the scenario which conditions the signal, it is driving factors which are prescribed in the scenario. And the scenarios are well thought off pathways of the future evolution of the climate change signal. This sentence here can be understood by climate critics that you only have to choose the right scenario to get the answer you want. This should not be written like this.**

**Response**: Thank you for pointing out this unclear formulation. We have rewritten the sentence in the revised manuscript (Page 2 Line 28f).

**Referee #3**: **Line 32f: what are the problems with open boundary conditions you mention here? This is one of the key features of your model setup so please describe it with more insight.**

**Response**: One of the main problems of AORCMs is the prescription of lateral boundary conditions for the regional ocean models which are mainly based on monthly means from global ocean reanalysis data sets (e.g. HYCOM [Metzger et al., 2014]), damping the ocean dynamics on time scales less than 1 month. Those regional climate models should effectively resolve the small-scale processes that are not adequately represented in the coarser model data used as boundary conditions. This creates inconsistencies between the regional model solution and the external data that can be avoided with the consideration of a global ocean model with refines resolution within the coupled domain (e. g. ROM) (Sein et al. 2015). Such an approach was employed by Izquierdo and Mikolajewicz (2019) in an ocean-only process study to account for the impact of the interaction of processes of different space and time scales on the Mediterranean Water Outflow (MOW) spreading, of particular importance in the Strait of Gibraltar and the Gulf of Cádiz. We have included this information in the revised manuscript (Page 3 Line 1f).

**Referee #3**: **Line 34: it is the first detailed evaluation but the run has also been used in other studies? A bit of a contradiction here.**

**Response**: Darmaraki et al. (2019) evaluated only the SST using the historical simulation (1976-2005, our ROM_P1) in a study that compares the performance of 17 simulations with 6 different models, therefore, the evaluation is only partial. This one is the first comprehensive evaluation of ROM high-resolution atmosphere-ocean simulation for present climate (1980-2012, ROM_P0).

**Referee #3**: **Page 3: line 4: evolution at the end? Maybe towards the end would be better. It is a process.**

**Response**: Thank you. We have made the correction in the revised manuscript.

**Referee #3**: **Line 6: skill not skills**

**Response**: We have made the correction in the revised manuscript.

**Referee #3**: **Line 8: better: driving model (mpi-esm) and skip the last part of the sentence.**

**Response**: We have made the correction in the revised manuscript.

**Referee #3**: **Line 24: information**

**Response**: We have made the correction in the revised manuscript.

**Referee #3**: **Page 4, line 27: "we make a dynamical downscaled of present time simulation". Please rewrite this sentence. Why did you force the ROM with MPI-ESM-LR? Later it is getting clear, that LR and MR differ very much although it is nowhere discuss why, so why this forcing?**

**Response**: Thank you for the remarks. We have rewritten the wrong sentence in an appropriate way in the revised manuscript (Page 5 Line 15f).

Regarding the reasons behind using MPI-ESM-LR as driving model, we have included the explanation at the end of section 2.3. We also refer to Jungclaus et al. (2013) and Stevens et al. (2013) for a detailed analysis of both configurations.

"We have used MPI-ESM-LR to force ROM in experiments ROM_P1 and ROM_P2 because -LR was used in a wider set of CMIP5 experiments and with more realizations than -MR (Giorgetta et al., 2013). Both present the same horizontal resolution in the atmosphere, and although MPI-ESM-MR has a higher vertical resolution, mainly in the upper troposphere and lower stratosphere, the main differences in the simulations can be found in the middle atmosphere (Stevens et al, 2013). According to a recent benchmarking exercise of CMIP5 models (Lauer et al., 2017) their overall performance is quite similar. Jungclaus et al. (2013) provided a detailed description and evaluation of the ocean performance of MPI-ESM-LR and -MR, and concluded that both behaved similarly in many aspects, although -LR simulated the Labrador Sea and the North Atlantic more accurately, at least in the mean state and mean variability features."

**Referee #3**: **Page 5, line 23: "we made comparisons". No good english.**

**Response**: Thank you. We have made the correction in the revised manuscript.

**Referee #3**: **Line 25: based on the NEMO code. Which model do you talk about here? Generally, I would skip a lot of information on the data but the resolution in space and time and the citations. But this is your choice.**

**Response**: We are talking about MEDSEA_REANALYSIS_PHY_006_009. The ocean global climate model used to produce the MEDSEA_REANALYSIS_PHY_006_009 are NEMO version 3.2 for the period 1955-2012 and NEMO version 3.4 for the period 2013-2015 (Fratianni et al., 2015). In any case we have followed your suggestion and deleted some information in the revised manuscript (Page 6 Line 24f).

**Referee #3**: **Line 30: against other ESMs are required.**

**Response**: Thank you. We have made the correction in the revised manuscript.

**Referee #3**: **Line 30: Why and how is the setup of MPI-OM different compared to MPI-ESM? And how are these differences relevant when looking at the results of ROM compared to the MPI-ESM-LR/MR?**

**Response**: The MPI-OM configuration used for all experiments features the grid over North America and Northwestern Africa. The horizontal resolution ranges from 5 km (close to the NW African coast) to 100 km in southern oceans (see Fig. 2b). This feature allows a local high resolution in the region of interest allowing the study of local-scale processes while maintaining a global domain (e.g. Izquierdo and Mikolajewicz, 2019). ROM-MPI-OM has higher resolution in the Mediterranean than any of the MPI-

ESMs. The low resolution (LR) configuration uses for the ocean a bipolar grid with 1.5° resolution and the medium resolution (MR) decreases the horizontal grid spacing of the ocean to 0.4° with a tripolar grid, two poles localized in Siberia and Canada and a third pole at the South Pole (Giorgetta et al., 2013).

The higher resolution will play a role in the ROM results.

**Referee #3: Page 6, line 1: is HAMOCC used? What role does it play in the MedSea climate change simulations? Why is OASIS3 used and not OASIS3-MCTx?**

**Response**: The HAMOCC model is coupled to our ROM coupled system, but we did not switch HAMOCC on in our runs. So HAMOCC is not accounted for in the simulations, and therefore does not play a role in our Mediterranean Sea climate change results. However, we think that switching HAMOCC on 1) would reduce the summer cold SST bias in ROM_P0 (see below our answer to one of your remarks), and potentially would have an impact on the Mediterranean climate change.

We are aware that OASIS3-MCTx offers better possibilities. However, the model was developed with OASIS3 and for now we are keeping it as the coupler. We are planning to change to OASIS3-MCTx in the near future, when we will run simulations with higher resolution. The limitations of OASIS3 will then most likely became critical.

**Referee #3: Line 17f: what is the explanation for the overestimation of the Azores high? The location is quite close to the boundary of REMO so where does it come from? The boundary formulation? If the ocean is the source, then there are deficits in the ocean circulation in the Atlantic which might play a role in the MedSea as well.**

**Response**: The REMO domain used in the simulation also includes the North Atlantic and the Azores high is included in the domain and its core is located far enough from the boundaries (the closer boundary is the northern one, see Fig. 2b). The biases in its representation (overestimation of MSLP) can be attributed to the model, in fact, to a SST cold bias in the North Atlantic. This feature is common to ocean GCMs and also appears in the MPI-OM configurations used in MPI-ESM-LR and -MR. This cold SST bias is related to the difficulties that the model has in accurately capturing the path of the Gulf Stream and North Atlantic Current (Keeley et al., 2012, Jungclaus et al., 2013). Our oceanic component has a high resolution, permitting eddy in the North Atlantic. However, we found that although the path of the simulated Gulf Stream and North Atlantic Current is improved compared to MPI_ESMs and most of the global coupled models, the improvement is not enough to completely alleviate the MSLP bias. As indicated in Sein et al. (2018), the resolution of the oceanic model plays a very important role, and significant improvements in the simulation of the Azores High can made using an ocean model that is eddy resolving in the region of the frontal currents.

**Referee #3: Line 23f: Could it be that REMO has a problem simulating the circulation over mountainous areas? There used to be a formulation in REMO smoothing out this effect, maybe this was not used? Generally, the great benefit of regional climate modelling is the higher resolution accompanied by a better representation of mountains. There is a sentence (Page 7, line 9f) in the paper talking about the benefits but leaving out this mountain/orography effect. So please discuss this point somewhere.**

**Response**: We indeed use the formulation that improves the representation of the circulation over mountainous regions in REMO (see Page 8 Line 5f). We think that in this case a significant part of the discrepancies arises from the differences in the height of the mountains in the model and reanalysis due to the different resolution of REMO (25 km) and ERA-Interim (c.a. 80 km; T255 spectral).

**Referee #3: Line 26: could play a role. Good question. Do they? Did you look at this point in more detail? Or is it due to interpolation artefacts?**

**Response**: The Air Temperature biases in this are mainly related to the SST biases. In winter, ROM's SST is warmer than OISSTV2 in this region, most likely due to the relatively coarse atmospheric resolution. For instance, Akhtar et al. (2018), found that a higher atmospheric resolution strengthens the simulated 10 wind, which increases the latent heat release and therefore lowers the SST. In our model

weaker than observed winds lead to a lower latent heat, warmer SST and therefore, lower air surface temperature.

**Referee #3: Line 30: Same is true for the last sentence on this page, did you check where the anomalies come from?**

**Response**: This can be related to the transport of precipitable water, which is influenced by the simulated evaporation over the ocean.

**Referee #3: Page 7, line 11ff: Why is MSLP higher in the coupled run?**

**Response**: The higher MSLP are mainly related to cold SST biases.

**Referee #3: Page 8, line 2: more the northern part of the eastern MedSea.**

**Response**: We have made the correction in the revised manuscript.

**Referee #3: Line 6: higher resolution where ocean or atmos?**

**Response**: Higher resolution in both.

**Referee #3: Line 8: what simulation is discussed here? P0?**

**Response**: Yes, it is ROM_P0. As we have said above, we have included the name of each simulation throughout the revised text.

**Referee #3: Line 17f: what data set configuration?**

**Response**: The lower horizontal resolution of EN4 v4.1.1 (1°x1°).

**Referee #3: Line 20: this is the first time, aerosols are mentioned. It would be nicer to have this information in the general description of the model. And then, why are aerosols neglected?**

**Response**: Thank you for the remark. The information on aerosols is now included in the REMO description section. Our version of REMO does not include an aerosol module. We use climatological values (Tanre et al., 1984) for aerosols and the longer time scale aerosol forcing is not represented. The aerosol climatology used has some deficiencies related to its low resolution and an unrealistic dust component that are reflected in the weakness of the increasing trend of modeled SST (see Page 4 Line 9f).

**Referee #3: Line 21: offset of SST: what about the spin-up of the ocean model?**

**Response**: See our answer to the 2$^{nd}$ referee #3 comment.

**Referee #3: Line 25f: last part of the sentence is not understandable. And what is climate uncertainty? Please define.**

**Response**: Thanks for the remark. Climatological was not the appropriate word. We think of the observational gridded datasets RMSE as a measure of the "observational" uncertainty. We have rewritten the sentence trying to express it in a better way (see Page 9 Line 24f).

**Referee #3: Line 27: Chapter 3.3: The influence of HD is nowhere discussed. Why is there so much freshwater inflow from HD? This is missing in this chapter.**

**Response**: Thank you for the remark. There is not so much freshwater inflow from HD. Indeed, HD freshwater inflow is below estimates from other authors (there was a mistake in the figure corresponding

to river runoff in Table 4, the correct value is 0.06 Sv). Now we have added a brief HD discussion to this chapter.

**Referee #3: Page 9, line 3: why is ROM always saltier? HD? Spin-up? Surface fluxes? Please explain.**

Response: ROM_P0 is closer to EN4 and CMEMS climatologies than MPI-ESM-LR and -MR simulations. This improvement in SSS is due to improved surface fluxes due to the higher horizontal resolution in the Mediterranean Sea and partly due to the seasonally varying fresh water flux correction.

**Referee #3: Line 29: RCSM4 model turns up out of the blue. What model is this, citation?**

Response: Thank you for the remark. We have included the citation for RCSM4 model (Sevault et al., 2014) in the revised manuscript (Page 10 Line 32).

**Referee #3: Page 10, first paragraph: global warming everywhere? Please skip such general sentences.**

Response: We agree with the reviewer. We have made the correction in the revised manuscript.

**Referee #3: Line 13: Figure 13 is not easy to understand, so please label this figures themselves with variable and time slot.**

Response: We have labeled all figures in the revised manuscript.

**Referee #3: Line 6f: which run is discussed, which period. And an av. Zonal SST from north to south is really hard to distinguish from that figure.**

Response: We are discussing the ROM_P1 simulation (1976-2005, Fig. 13a). We have rephrased the sentence.

**Referee #3: Line 21: already in the present day simulation is too much fresh water in the Bosporus and close to the Nile, this will be transported to the future scenarios.**

Response: The Reviewer is right. There is an excess of fresh water in those areas due to an excessive fresh water input from Dardanelles and from the Nile. However, this effect is quite localized. This pattern is also evident in ROM_P1, so it seems that it can be transported into future scenarios. However, its impact would be quite local and restricted to the northern Aegean Sea and close to the Nile. The freshening of the western Mediterranean is clearly related to the Atlantic influence: see Fig. 15, where Western Mediterranean (averaged) gets fresher at the surface, while the Eastern Mediterranean gets saltier.

**Referee #3: Line 26f: where does the diff between MR and LR come from?**

Response: Differences are related to different ocean grids, a bipolar curvilinear grid (GR1.5, nominal 1.5º resolution) for -LR and a tripolar curvilinear grid (TP04, nominal 0.4º resolution) for -MR (Jungclaus et al., 2013).

**Referee #3: Line 19: where did you discuss the circulation in the Adriatic Sea and compared it to observations?**

Response: We have not discussed the circulation in the Adriatic Sea. We wanted to stress ROM's ability to reproduce smaller scale details. The sentence is rewritten in the revised version.

**Referee #3: Page 12, first paragraph: the discussion would be a bit more interesting if the resolution of the different models mentioned compared to ROM would be written somewhere.**

**Response**: We have included a new Table 5 where the horizontal resolution of the different models has been summarized.

**Referee #3: Line 27f: so why do you speculate and not do some analysis and answer this question**?

**Response**: We have added a piece of text describing a preliminary analysis on this issue:

"It is difficult to attribute the bias to single cause without considering the multiplicity and complexity of all the involved conditions; therefore, this topic deserves a separate and focussed study. However, a preliminary sensitivity analysis changing the optical properties of the water (changing from model standard Jerlov Ia to Jerlov II) clearly indicates that the related turbidity increase is responsible for a larger absorption of downward shortwave radiation in the upper layer, and leading to a warmer SST. This also would explain why colder SST biases appear in summer, when the impact of biologically-induced redistribution of heat in the water column is larger. Switching HAMOCC on would, to some extent, contribute to the reduction of this cold bias. However, until a thorough study is carried out, the contribution of other mechanisms cannot be discarded." (Page 13 Line 30f). Fig. 1 shows the results of the sensitivity experiment.

[Figure]

Fig. 1. DFJ and JJA SST differences, averaged for the period 1980-2012, between ROM_JerlovII and ROM_JerlovIa (Figure not included in the manuscript)

**Referee #3: Page 13, line 4: please mention the 31m in the model description. Is this the depth of the first layer or only the choice to compare it to other studies? Unclear.**

Response: The first layer depth of the model is 8 m. We have chosen the 31 m level depth to remove the high-frequency variability of the uppermost ocean while retaining a characteristic upper ocean circulation pattern. Also, the choice of this level depth makes our results more comparable with other studies (see Page 14 Line 15f). We have included information about the first layer depth in the model description (see Page 5 Lines 1-2).

**Referee #3: Line 11: RCSM4 is not to be found in Fig 12b.**

Response: Yes, you are right. We have not used RCSM4 model data in our study, we have just compared our result with those obtained by Sevault et al. (2014) with the RCSM4 model as we have carried out the same analysis.

**Referee #3:  Line 17: what is the absence of resolution? Sound more like a philosophical question.**

Response: We mean the lack of resolution. We have made the correction in the revised manuscript.

**Referee #3:  Line 24: what is the multi-model Mediterranean? And what are model increases? Lot of errors on this page.⊗Like differences instead of different and filed instead of field and so on. Not very nice to read. Lots more.**

Response: The English has now been checked by a professional.

**Referee #3: Page 14, line 9: where was the transport through the Dardanelles Strait discussed?**

Response: We have not discussed the transport through the Dardanelles Strait, although it was with available observation (Sanchez-Gomez et al., 2011) and RCMS4 model estimates (Sevault et al., 2014) (see Table 4). However, our point here is to indicate that the higher ocean resolution allows to explicitly resolve the water exchange through more realistic (in the sense of cross-sectional dimensions) straits.

**Referee #3: Generally: in the conclusion: I could like to see a good overview about the benefits of this coupled system ROM compared to others without the global ocean, a short overview over the evaluation, i.e. how well is the model behaving in the ERA-Interim simulation to have a good feeling about the future projections. And then the climate change signal which was simulated. Maybe some problems/shortcomings of the model and some ideas how to improve well known deficits.**

Response: We have substantially updated the discussion and conclusions (sections 4 and 5) in the revised manuscript.

**List of relevant changes:**

- We have fully reviewed the manuscript following the comments and the suggestions that the reviewers have done. We have tried to answer all the referee's questions which were full of "whys" in the revised manuscript such as:

- why we use MPI-ESM-LR as driving model instead of -MR
- how the spin-up of the ocean model have been calculated
- description of ROM configuration and experiments set-up
- differences between MPI-OM and MPI-ESM
- the role of HAMOCC over the STT bias
- explanation of overestimation for the Azores high
- why are aerosols neglected
- the influence of HD

Furthermore, we have substantially updated the discussion and conclusions, where we have included more insight analysis and a good overview about the benefits of this coupled system model. You could find all changes in the marked-up manuscript version bellow.

- We have used the names of each ROM simulations (P0, P1, P2) and labeled the figures throughout the revised manuscript.

-We have re-plotted the Fig. 11 choosing 0.1 m/s as scaling factor to improve the figure.

- Water budget calculation was wrong because there was a mistake in providing the values for river run-off (R). Now the results are checked and the water budget is ok.

[revised manuscript text omitted]